# LIME: Making LLM Data More Efficient with Linguistic Metadata Embeddings

## Abstract

Pre-training decoder-only language models relies on vast amounts of high-quality data, yet the availability of such data is increasingly reaching its limits. While metadata is commonly used to create and curate these datasets, its potential as a direct training signal remains under-explored. We challenge this status quo and propose LIME (**Li**nguistic **M**etadata **E**mbeddings), a method that enriches token embeddings with metadata capturing syntax, semantics, and contextual properties. LIME substantially improves pre-training efficiency. Specifically, it adapts up to 56% faster to the training data distribution, while introducing only 0.01% additional parameters at negligible compute overhead. Beyond efficiency, LIME improves tokenization, leading to remarkably stronger language modeling capabilities and generative task performance. These benefits persist across model scales (500M to 2B). In addition, we develop a variant with shifted metadata, LIME$^{+1}$, that can guide token generation. Given prior metadata for the next token, LIME$^{+1}$ improves reasoning performance by up to 38% and arithmetic accuracy by up to 35%.

## 1 Introduction

Autoregressive language models have emerged as a prominent area of research due to their impressive capabilities. However, training these large language models (LLMs) is computationally expensive and highly data-intensive. Smaller LLMs are particularly attractive because of their reduced resource requirements and accessibility. Nonetheless, models up to 2B parameters require the same—or even increased—amount of training data, as their language modeling performance tends to regress (Hoffmann et al., 2022).

At the same time, the availability of novel human-generated high-quality training data is decreasing (Xue et al., 2023; Villalobos et al., 2024), emphasizing the need of improving the utility of existing datasets. To compensate this shortcoming, methods of accumulating LLM pre-training datasets shift from mere quality filtering to synthetization through earlier model generations and staging of increasing data quality buckets (Su et al., 2025). In order to determine the stage and quality bucket, existing document-level metadata is used, along with more complex—and even model-based—scores that reflect attributes such as educational value, or factual reliability (Schuhmann et al., 2022; Li et al., 2024; Penedo et al., 2024; Wettig et al., 2025).

However, neither pre-existing nor created metadata are typically propagated downstream into the model during training. Modern LLM tokenizers are typically trained on yet another blended dataset with solely text compression as objective, neglecting linguistic research entirely. As such, tokenization can fragment meaningful content, distort sequence relationships, and ultimately degrade the efficiency and quality of model learning. Recent work indicate that linguistically motivated segmentation can improve model training (Hou et al., 2023; Schmidt et al., 2024). Moreover, early work suggests that linguistic token annotation can improve certain modeling capabilities such as in machine translation (Sennrich & Haddow, 2016).

To this end, we introduce a method which rigorously integrates token-grained linguistic metadata into LLM pre-training at negligible complexity and computational overhead: LIME (Linguistic Metadata Embeddings for LLMs). Our method augments pre-trained subword tokenizer with linguistically informed annotations, namely POS and NER tags, extending its standard output of raw tokens with token-aligned metadata. We propagate metadata downstream by incorporating it as additional input

signals to shift the token embedding space. As a guidance variant, with `LIME`$^{+1}$ we shift the metadata embeddings by one to guide the generation with look-ahead metadata.

In our experiments, we demonstrate that `LIME` substantially enhances language modeling performance. By incorporating metadata, `LIME` improves data efficiency during training, enabling models to adapt up to 56% faster to the training data distribution. Additionally, `LIME` mitigates issues caused by artificial tokenization splits, keeping the meaning of subword tokens together, as supported by both qualitative and quantitative analyses. Finally, with `LIME`$^{+1}$ we demonstrate that models achieve up to 35% higher accuracy when the metadata class of the token to be predicted is revealed in advance. Crucially, we apply this guidance in tasks such as for reasoning and arithmetic, where the relevant metadata is naturally available rather than artificially constructed. These benefits of metadata annotations persist consistently throughout our scaling ablations of 500M, 1B and 2B parameter models. Our results raise important questions about inefficient pre-training data usage in standard causal language model training.

Our main contributions and findings are summarized as follows:

1. We introduce `LIME` and `LIME`$^{+1}$, our approach to augment token embeddings with linguistic metadata in Sec. 3.

2. We demonstrate how `LIME` improves language modeling capabilities, in particular next-token prediction, consistently across various model sizes (Sec. 4.2).

3. `LIME` models excel in generative downstream tasks (Sec. 4.3) and keep split word tokens together by improving natural language word cohesion (Sec. 4.4).

4. `LIME`$^{+1}$ enables inference-time metadata steering which improves reasoning and arithmetic capabilities (Sec. 4.5).

## 2 RELATED WORK

Before introducing `LIME`, we outline key areas of prior work that motivate and inform our approach. Specifically, we review tokenization strategies and their implications for model efficiency, the role of metadata in LLM pre-training, the use of linguistic annotations as auxiliary supervision, and recent efforts to integrate metadata directly at the embedding level.

**Pre-Tokenization and Tokenizers.** Pre-tokenization defines segment boundaries for subword tokenization by normalizing and splitting text (e.g., on whitespace or punctuation) into coherent units. Subword tokenizers, trained with compression-based methods like BPE (Sennrich et al., 2016), inherit biases from their training data: Ahia et al. (2023) report large cross-lingual disparities, with some languages requiring up to five times more tokens for the same content. Tokenizers optimized for one distribution may become inefficient under distribution shifts (Ahia et al., 2023; Deiseroth et al., 2024; Neitemeier et al., 2025). Thus, fragmentation, or more tokens per word, correlates with poorer model performance. Linguistically informed segmentation can improve results: Hou et al. (2023) find morphological splits reduce perplexity and maintain or improve downstream accuracy, while Schmidt et al. (2024) show ignoring morphology in pre-tokenization can harm performance. Recent work explores byte-level or tokenizer-free models such as ByT5 (Xue et al., 2023) and MegaByte (Yu et al., 2023), and T-FREE (Deiseroth et al., 2024), which embeds words via character trigrams, capturing morphological overlaps with smaller embeddings.

**Metadata in LLM Pre-training.** Pre-training refers to an LLM learning from scratch on large corpora to establish a foundation for downstream adaptation. LLM downstream performance is strongly influenced by the quality of pre-training data (Longpre et al., 2024; Wettig et al., 2024). To improve quality, pre-training data is filtered and deduplicated using metadata typically derived from heuristic approaches (Raffel et al., 2020; Rae et al., 2021) or model-based classifiers (Brown et al., 2020; Xie et al., 2023; Penedo et al., 2024; Li et al., 2024; Su et al., 2025). Further, several approaches have been proposed that, instead of leveraging metadata solely to improve data quality, propagate metadata directly into model training. For example, CTRL (Keskar et al., 2019) prepends source-domain metadata, Dhingra et al. (2022) prepend timestamps to improve memorization, Liu et al. (2020) add language identifiers for multilingual training, and Khalifa et al. (2024) include document identifiers to improve source attribution. Most recently, Allen-Zhu & Li (2025) demonstrated that prepending a special token to useful data significantly increases the model's capacity ratio,

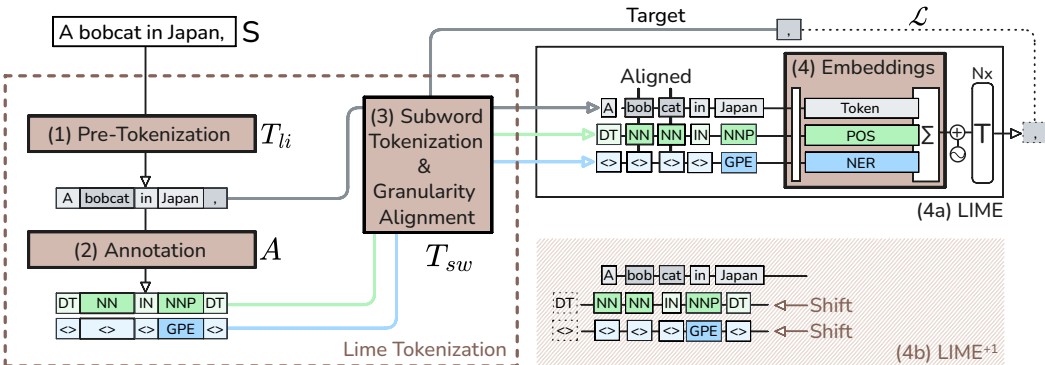

Figure 1: LIME and LIME$^{+1}$ architecture. **(1)** Input text S is split by the linguistic tokenizer ($T_{li}$). **(2)** Linguistic splits are annotated, e.g. with POS and NER tags. **(3)** Subword tokenization ($T_{sw}$) is applied to the linguistic tokens and annotations are aligned to the new splits. **(4)** Tokens and metadata are embedded, fused together and passed into consecutive transformer blocks.

while Gao et al. (2025) provided empirical evidence by showing a 33% improvement in pre-training efficiency when URL metadata was prepended. While effective, these methods rely on the existing vocabulary to encode metadata, which consumes valuable input token space, and they typically operate at the document level, limiting annotation granularity.

**Auxiliary Supervision with Linguistic Annotations.** Incorporating fine-grained linguistic token annotations into neural NLP models has been widely studied. For example, Sennrich & Haddow (2016) showed that features such as POS tags and dependency relations can improve neural machine translation. These ideas have since been extended to pre-training: ERNIE (Sun et al., 2021) leverages entity-aware masking guided using NER, while syntax-aware pre-training models incorporate dependency structures into attention mechanisms (Zhang et al., 2022). Knowledge-enhanced pre-trained language models (KEPLMs) (Hu et al., 2024) integrate structured information, such as knowledge graphs. Yet, these approaches substantially increase architectural complexity and computational cost.

**Embedding-Level Metadata Integration.** Embedding layers serve as an effective mechanism for metadata injection, enabling structured information to be incorporated directly into the model's representational space. For instance, (Guu et al., 2020) propose retrieval-augmented pre-training using a knowledge-enriched encoder. The joint learning of language and knowledge embeddings within a masked language modeling framework has been investigated by Sun et al. (2020). CUE (Novotney et al., 2022) incorporates metadata, such as author and date, into LLMs through a separate context encoder. Furthermore, McLeish et al. (2024) showed that embedding positional cues into the representations of numerical tokens can enhance performance on arithmetic tasks. Armengol-Estapé et al. (2021) investigate the extension of token embeddings with linguistic information to improve low-resource machine translation in bidirectional transformer models. Nevertheless, incorporating fine-grained linguistic metadata into the token embeddings of autoregressive LLMs remains an open research direction. We address this gap with LIME, showing that it delivers scalable improvements in both efficiency and performance.

## 3 LIME: LINGUISTIC METADATA FOR LLMS

In this section, we present LIME which integrates linguistic metadata for LLMs. LIME consists of four stages: (1) linguistic pre-tokenization, (2) metadata annotation, (3) subword tokenization & granularity alignment, (4) and metadata embedding, as shown in Fig. 1.

**Illustrative Example.** Before introducing technical details, we summarize LIME($^{+1}$). The method enriches input text with linguistic metadata (e.g., POS or NER tags) before passing it to an LLM. As illustrated in Fig. 1, a sentence is first split into linguistic tokens (e.g., A bobcat in Japan → four tokens), which are then annotated with metadata such as POS tags. Since LLMs operate on subword units, the metadata is aligned with the subword tokenizer output to preserve dimensions. The tokens and metadata are then embedded, combined, and fed into the transformer. LIME applies each token's

Table 1: Illustrating `LIME` and `LIME`$^{+1}$ with the example of Fig. 1. { } denote embeddings.

| (0) | Input Sequence | A␣bobcat␣in␣Japan | | | | |
|---|---|---|---|---|---|---|
| **(1)** | Linguistic Pre-Tokenization | `A` | `␣bobcat` | | `␣in` | `␣Japan` |
| **(2)** | Metadata Annotation | `DT` | `NN` | | `IN` | `NNP` |
| **(3)** | Subword Tokenization & Granularity Alignment | `A` `DT` | `␣bob` `NN` | `cat` `NN` | `␣in` `IN` | `␣Japan` `NNP` |
| **(4a)** | Metadata Embeddings `LIME` | { `A` } +{ `DT` } | { `␣bob` } +{ `NN` } | { `cat` } +{ `NN` } | { `␣in` } +{ `IN` } | { `␣Japan` } +{ `NNP` } |
| **(4b)** | Metadata Embeddings `LIME`$^{+1}$ | { `A` } +{ `NN` } | { `␣bob` } +{ `NN` } | { `cat` } +{ `IN` } | { `␣in` } +{ `NNP` } | { `␣Japan` } +{ `X` } |

own metadata, while `LIME`$^{+1}$ shifts embeddings to use the next token's metadata, giving the model richer linguistic context. These stages are further illustrated in Tab. 1.

**(1) Linguistic Pre-Tokenization.** We define a tokenizer on the alphabet $\Sigma$ as the tuple $T = (f, V)$ consisting of the function $f : \Sigma^* \to V^*$ that splits a given text $s \in \Sigma^*$ into a sequence of tokens of the token vocabulary $V$ (Minixhofer et al., 2024). A token sequence is denoted as $T(s) = t_1, t_2, ..., t_n$ of length $n$. Unlike the conventional approach of using only a statistically learned subword tokenizer $T_{sw}$, we introduce a rule-based linguistic tokenizer $T_{li}$ for pre-tokenization. Introducing $T_{li}$ allows for effective and linguistically-informed segmentation into minimal meaningful text units, enabling the assignment of fine-grained metadata labels.

**(2) Metadata Annotation.** The metadata annotation process is defined by an annotator $A = (g, C)$ with the annotation function $g : V^n \to C^n$ that, for a given token sequence $T(s)$ of length $n$ produces an annotation sequence $A(T(s)) = a_1, a_2, ..., a_n$ with $a \in C$ and $C$ being the set of pre-defined annotation symbols. The annotation function $g$ can consist of rule-based methods, heuristics or classification models. As illustrated in Fig. 1, our method allows to define and integrate multiple annotators, e.g., POS and NER annotations.

**(3) Subword Tokenization & Granularity Alignment.** Given the subsequent subword tokenizer $T_{sw} = (f_{sw}, V_{sw})$, its vocabulary and annotation function will naturally differ from those of the rule-based word tokenizer $T_{li}$ used in the first stage, as shown in Fig. 1. This entails that $n_{li} = |T_{li}(s)| \neq |T_{sw}(s)| = n_{sw}$ has to be aligned for an input text $s$. In the case of $n_{li} < n_{sw}$, we define the annotation function $g' : V^{n_{li}} \to C^{n_{sw}}$ that resolves this granularity mismatch as follows: $g'$ tracks for every token in $T_{sw}(s)$ its word context defined by $T_{li}(s)$ and repeats the relevant annotation until a granularity match is achieved. In the case of $n_{li} > n_{sw}$, where $T_{li}$ splits a input sequence in more tokens than $T_{sw}$, we keep the finer granularity of $T_{li}$ by expressing every token in $T_{li}(s)$ with tokens from the vocabulary $V_{sw}$. For example, the word "don't" could linguistically be split as $T_{li}(s) = $ `do` `n` `'` `t` while the subword tokenizer may produce $T_{sw} = $ `don` `'` `t`. In that case we keep the slightly less compressed word boundaries of $T_{li}(s)$ and their corresponding annotations as the subsequent tokenization of $T_{sw}$, to not lose annotation precision. Now, for a set of metadata domains $D$, and $|D|$ metadata annotators, the final output of this stage is a granularity aligned sequence consisting of tuple $(t_i, (a_{i,d})_{d \in D})$ for token index $i$. We refer to these first three stages as `LIME Tokenization`.

**(4) Metadata Embeddings.** In our output tuple $(t_i, (a_{i,d})_{d \in D})$ at index $i$, token $t_i$ is one-hot encoded to a vector of length $|V_{sw}|$ and multiplied with the embedding matrix $W^L \in \mathbb{R}^{|V_{sw}| \times h}$, $h$ being the hidden size hyperparameter, producing the language token embedding $E_L(t_i)$. From here we introduce two variants to blend in token annotations. The first, termed `LIME` (see Fig. 1 top), uses metadata embeddings $E_M^d$ originating from the same token:

$$E(t_i) = E_L(t_i) + \sum_{d \in D} w_d E_M^d(a_{i,d}) \ . \tag{1}$$

Note that the annotation embedding process is applied to individual respective matrices $W^{(d)} \in \mathbb{R}^{|C_d| \times h}$ producing $E_M^d(a_{i,d})$. $E_L$ combined with the weighted sum of $|D|$ metadata embeddings creates the final embedding and LLM input $E(t_i)$.

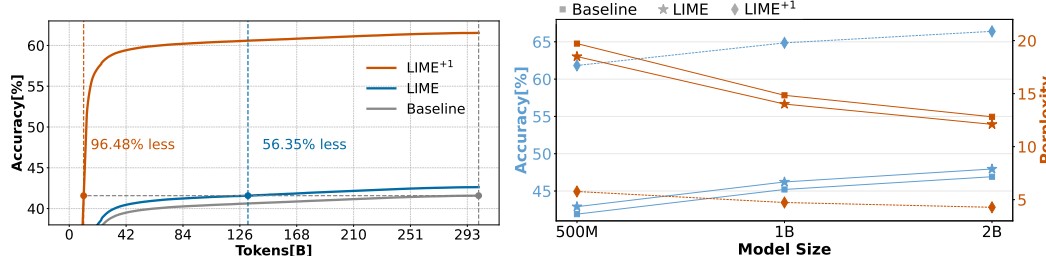

Figure 2: **Left:** Next-token accuracy improves with metadata embedding layers. Our `LIME`500M model requires 56% less pre-training data to achieve the same token prediction accuracy as `Baseline`. **Right:** Accuracy and perplexity improvements translate consistently across model sizes.

The second implementation (see Fig. 1 bottom), termed `LIME`[+1], is tailored for scenarios where the metadata annotation of the next token ($a_{i+1}$) is known in advance, such as demonstrated in Sec. 4.5. During training and inference (Fig. 4), the base embedding $E_L$ is augmented using the metadata embeddings of the *next* token, rather than the current one. This look-ahead embedding is defined as:

$$E(t_i) = E_L(t_i) + \sum_{d \in D} w_d E_M^d(a_{i+1,d}) .$$ (2)

Note, that we modify only the token embeddings; all other transformer components, including the LLM head and the cross-entropy loss applied during training, remain unaltered and fully agnostic to the metadata.

## 4 EXPERIMENTS

We now present empirical evidence demonstrating the benefits of `LIME` and `LIME`[+1]. We start with the experimental setup followed by an investigation of model prediction qualities (Sec. 4.2) and evaluations on popular benchmarks (Sec. 4.3). We then provide qualitative and quantitative analyses that `LIME` models improve token coupling (Sec. 4.4) and finally show that `LIME`[+1] excels in reasoning as well as arithmetic performance (Sec. 4.5).

### 4.1 EXPERIMENTAL SETUP

In our experiments, we followed the Gemma architecture (Mesnard et al., 2024) and pre-trained `LIME` models in three sizes, 500M, 1B, 2B, on 302 billion tokens of the `DCLM-BASELINE` (Li et al., 2024) dataset (cf. App. A.2). We applied the three-stage `LIME Tokenization` process described in Sec. 3, with $T_{li}$ being the english spaCy tokenizer[1] and $T_{sw}$ the SentencePiece (Kudo & Richardson, 2018) Gemma tokenizer, and otherwise optimizing the standard cross-entropy loss $\mathcal{L}$ on the language head (disregarding metadata). As mentioned in Sec. 3, extending tokenization with $T_{li}$ increases token count, in our case by 1.19%, but compresses the used vocabulary by 1.03%, and as such frees embedding parameters (cf. App. A.3). Models trained with `Lime Tokenization` but no additional metadata embedding layers are referred to as `Baseline` in the following.

We extended the `Baseline` models with two metadata domains available in spaCy: Syntactic Part of Speech (POS), with $|C_{\text{pos}}| = 51$ (App. A.11), and semantic Named Entity Recognition (NER), with $|C_{\text{ner}}| = 20$ (App. A.12). Both annotation embedding layers are weighted equally with $w_{\text{pos}} = w_{\text{ner}} = 1$ (Eq. 1). For both `LIME` and `LIME`[+1], the additional embedding layers, having 71 entries in total, add less than $0.01\%$ to the total parameters. Since it requires only one vectorized lookup and addition in the forward pass, it adds negligible runtime overhead. For details, see App. A.3. `LIME`([+1]) inference is illustrated in Fig. 4.

---

[1] https://spacy.io/api/tokenizer

Table 2: LIME excels at generative tasks. Improvements to Base are indicated by ↑, to LIME with ⇑. We highlight (yellow) generative-format tasks. Exemplified on 500M, other model sizes in App. A.5.

| | Base$_{500M}$ | LIME$_{500M}$ | LIME$^{+1}_{500M}$ |
|---|---|---|---|
| ARC-Easy [7] | 57.00±1.57 | ↑57.20±1.57 | ⇑58.10±1.56 |
| BoolQ [6] | 49.50±1.58 | 47.90±1.58 | ⇑59.40±1.55 |
| COPA [35] | 62.00±4.88 | ↑64.00±4.82 | 61.00±4.90 |
| HellaSwag [51] | 36.20±1.52 | ↑37.00±1.53 | ⇑43.10±1.57 |
| LAMBADA [30] | 26.50±1.40 | ↑29.80±1.45 | ⇑49.00±1.58 |
| OpenBookQA [26] | 31.00±2.07 | ↑32.60±2.10 | ⇑34.20±2.12 |
| PIQA [4] | 69.90±1.45 | 69.40±1.46 | 68.20±1.47 |
| TriviaQA [16] | 8.20±0.87 | ↑ 9.80±0.94 | ⇑19.50±1.25 |
| WinoGrande [36] | 51.70±1.58 | ↑52.60±1.58 | ⇑54.60±1.58 |
| Mean | 43.56±1.88 | ↑44.48±1.89 | ⇑49.68±1.95 |

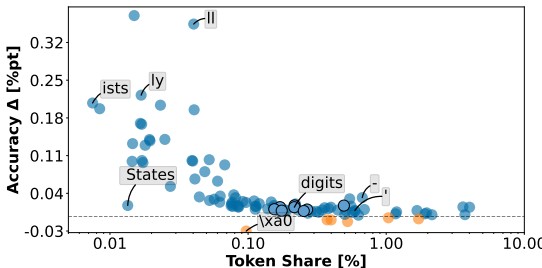

Figure 3: LIME$_{500M}$ token prediction accuracy increases within semantic and syntactic metadata class boundaries: Among the 100 most impactful (share×Δ) tokens, our model exhibits improved token coupling for suffix-, digit-tokens and an exemplary entity group token ␣States.

## 4.2 LINGUISTIC METADATA-EMBEDDINGS IMPROVE LANGUAGE MODELING

We first analyze the effect of metadata embeddings on general pre-training dynamics. Across all three model sizes, LIME variants reach the same next-token accuracy as their baselines earlier in training. Fig. 2 (left) illustrates these gains for the 500M model. First, we observe that LIME$_{500M}$ achieves the final accuracy of its Baseline counterpart (41.90%) using 56.35% fewer tokens. Specifically, both models follow the expected pre-training pattern of rapid early gains. For 500M, gains begin plateauing after about 42B tokens. From that stage onward, LIME maintains a stable accuracy advantage, suggesting that extended pre-training of the baseline cannot substitute for LIME metadata embeddings. This training dynamic is consistent across all three model sizes.

Further, we evaluate model quality across sizes on a DCLM-BASELINE test set of 10,000 samples, reporting next-token accuracy and perplexity on the same batches of data (Fig. 2 right). We observe that all LIME models have improved accuracy and reduced perplexity relative to their Baseline models. Accuracy increases by a constant magnitude of approximately +1 percentage points across all model sizes. Perplexity reduction is, as expected in this range, more pronounced in smaller model sizes. LIME$^{+1}$ shows a substantial improvement in accuracy and perplexity. Relative to Baseline models, accuracy increases by over 40% across model sizes, while perplexity is reduced by more than 65%, enabling our LIME$^{+1}_{2B}$ model to achieve a perplexity of 4.30. Training was stable across all methods and model sizes. For more detailed results and learning curves, please refer to App. A.4. Next, we demonstrate that these results directly translate to downstream task performance.

## 4.3 LIME MODELS EXCEL IN GENERATIVE TASKS

We evaluate downstream transfer on standard benchmarks (Tab. 2), highlighting results for the 500M model; comparable trends hold for all sizes, detailed in App. A.5. Each task was run with 1,000 samples using randomly selected 5-shot contexts.

The results show that LIME models on average slightly improve over their respective Baseline models. These LIME improvements appear modest due to the use of multiple-choice logit comparisons, which may not fully capture the LIME benefits. On tasks that are evaluated using generative greedy sampling (e.g., LAMBADA/TriviaQA), our method improves model performance substantially across sizes.[2] Specifically, for our 500M model, LAMBADA increased from 26.50 to 29.80 and TriviaQA from 8.20 to 9.80. LIME$^{+1}$ models, on the other hand, improve in 500M and 1B on 7 of 9 tasks, and on 5 tasks in 2B parameters, when being compared to their respective Baseline and LIME. The substantial and constant improvement of LIME models on greedily sampled tasks is further amplified by LIME$^{+1}$ models: For TriviaQA, we observe a relative improvement of 138%, 90%, and 64%, and in LAMBADA of 85%, 61%, 53% for 500M, 1B, 2B, compared to their respective Baseline.

---

[2]The remaining tasks are multiple-choice log-likelihood evaluations compared to argmax matching.

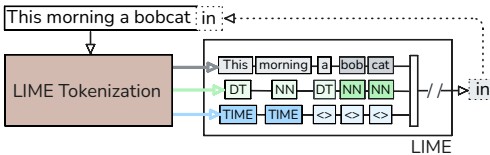 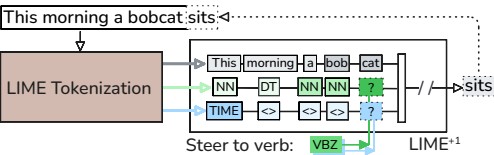

(a) LIME models receive metadata augmented input signal via LIME Tokenization for auto-regressive token prediction.

(b) LIME⁺¹ models can steer in metadata directions. Shifting metadata by one and injecting VBZ and <> for ⎵cat steers the model to generate a verb token next.

Figure 4: Inference with LIME and LIME⁺¹ models.

## 4.4 LIME KEEPS TOKENS TOGETHER

Building on the previous finding that LIME boosts performance in generative tasks, we aim to better understand where and how these improvements arise at the token level. To this end, we first conduct a qualitative study of token-level accuracy shifts, followed by a detailed quantitative analysis across word lengths, model sizes, and architectures.

**Suffix Tokens, Entity-Groups & Digits.**   In the following, we take the dataset of Sec. 4.2 and rank all unique tokens by their shift in prediction accuracy and weight them by their share in the set. Illustrating accuracy shifts of the 100 most impactful tokens (Fig. 3) reveals: Suffix tokens such as ing and ly, that finish words consisting of multiple tokens, exhibit large positive accuracy shifts. This holds moreover for apostrophe and hyphen, as well as entity-group tokens, i.e. consecutive tokens that share the same entity metadata class, such as ⎵States from "United States". Finally, we observe a constant accuracy improvement across all single-digit tokens 0-9. Accumulating all positive and negative accuracy shifts resembles the total accuracy gain of LIME₅₀₀ₘ as reported in Sec. 4.2 (more details in App. A.6).

**Keeping Natural Language Word Tokens Together.**   Additionally, in Tab. 3 (left), we count occurring words of certain token lengths and show their averaged accuracies.[3] We observe that LIME models outperform their baselines across all cases, and notably, for word lengths $x \geq 2$.[4] For single-word tokens ($x = 1$), we still obtain gains of roughly 0.7%. This highlights that metadata embeddings not only have a strong effect on the cohesion of subword sequences, but also help improving in-context relevance.  LIME⁺¹ models further amplify the previously described improvements.  As expected, the look-ahead metadata of LIME⁺¹ compellingly improve single-word token prediction by +21%. Moreover, models trained that way even further double the token-coupling accuracies improvements to +10%. These findings underline to what extent certainty on token metadata can still improve next-token prediction, even on models with up to 2B parameters. More details in App. A.6.

## 4.5 METADATA GUIDANCE UNLOCKS HIDDEN REASONING AND ARITHMETIC ABILITIES

Finally, we demonstrate the impact of token-metadata on two common tasks, reasoning (FLenQA) and symbolic addition arithmetic (ARI-ADD).

**Task Definition.** FLenQA (Levy et al., 2024) requires reasoning across multiple, increasingly noisy contexts. As such it provides a measure of a model's ability to robustly handle reasoning scenarios. We use the FLenQA task 'People in Rooms' (PIR) that prompts to combine two pieces of information found in a given noisy context. Both facts are required to answer the question, as in the following example: *"John's room is blue walled [...] Ethan is in John's room [...] Is Ethan in a blue walled room?"* To evaluate our models that are trained on DCLM-BASELINE only, we convert the questions into the following generative format: *"John's room is blue walled [...] Ethan is in John's room [...] Ethan is in a room. It has the following properties:"* and greedily sample for the ground truth *" blue walled"* to appear in the subsequent words. Note that using LIME⁺¹₂ᴮ, we syntactically steer (Fig. 4b) the model to, e.g., in the prior example, predict an adjective token (JJ and <>) by overloading the last

---

[3]Details are found in App. A.6.

[4]Note that, the category of tokens in words of $x \geq 2$ accounts for 10% of the number of single-word tokens, since we apply an English-optimized tokenizer to a primarily English corpus.

Table 3: **Left:** `LIME` models have improved prediction accuracies on tokens within natural language words of length $x \geq 2$ (in tokens) and slightly improve on words consisting of one token ($x = 1$). $n$ denotes the total token count of the respective length. **Right:** Reasoning and arithmetic capabilities are significantly improved with $\text{LIME}_{2B}^{+1}$. Evaluated on FLenQA and ARI-ADD, an addition task. More details in App. A.7.

| size | $x$ | $n$ | Base | LIME | $\text{LIME}^{+1}$ | Task | $\text{Base}_{2B}$ | $\text{LIME}_{2B}$ | $\text{LIME}_{2B}^{+1}$ |
|------|-----|-----|------|------|--------------------|------|--------------------|--------------------|-------------------------|
| 500M | $\geq 2$ | 424,960 | 69.68 | 73.95 ↑ 4.27 | **80.41** ↑ 10.73 | FLQA-250 | 42.0 | 52.0 ↑ 10.0 | **80.0** ↑ 38.0 |
| 1B | $\geq 2$ | 424,960 | 73.32 | 77.48 ↑ 4.16 | **83.43** ↑ 10.11 | FLQA-500 | 49.5 | 65.3 ↑ 15.8 | **73.5** ↑ 24.0 |
| 2B | $\geq 2$ | 424,960 | 75.51 | 79.24 ↑ 3.73 | **84.95** ↑ 9.44 | FLQA-1000 | 34.8 | 47.0 ↑ 12.2 | **65.3** ↑ 30.5 |
| 500M | 1 | 5,049,553 | 33.91 | 34.59 ↑ 0.68 | **55.10** ↑ 21.19 | FLQA-2000 | 40.3 | 44.3 ↑ 4.0 | **52.5** ↑ 12.2 |
| 1B | 1 | 5,049,553 | 37.23 | 37.89 ↑ 0.65 | **58.35** ↑ 21.12 | FLQA-3000 | 28.2 | 30.0 ↑ 1.8 | **39.3** ↑ 11.1 |
| 2B | 1 | 5,049,553 | 39.08 | 39.80 ↑ 0.72 | **59.94** ↑ 20.86 | ARI-ADD | 22.6 | 26.9 ↑ 4.3 | **58.7** ↑ 36.1 |

token (`a`) of the context. The benchmark is grouped into 5 noise levels, adding irrelevant information of 250, 500, 1000, 2000, and 3000 tokens, and each group consisting of 400 samples.

Second, for the arithmetic task, we again try to leverage the already obtained arithmetic performance from `DCLM-BASELINE` without further finetuning. We found the prompt *"The result is: {number}+{number} = "* performs best. The greedily sampled completion is then compared with the correct numerical result, thus testing basic symbolic arithmetic capabilities in a generative setting. We randomly sample $5 \leq \{number\} \leq 49$ and average the results over 500 unique number pairs.

**Results.** In Tab. 3 (right), we report performances across the different FLenQA variants and the ARI-ADD task. For all tasks we only show the 2B accuracies, as the other model sizes further degenerated. Nevertheless, the described behavior is consistent across sizes and found in App. A.7 for completion.

In FLenQA we first observe a clear trend of decreasing performance with increasing noise context across all models, as it is expected. `LIME` already yields notable improvements over `Baseline`, especially for shorter contexts (e.g., +15.8 on FLQA-500). This again indicates that enriching the embedding with metadata is already helpful to improve next-token accuracy. In contrast, $\text{LIME}^{+1}$ consistently delivers two-digit improvements across all variants, still achieving 39.3% on FLQA-3000 and improving even 38% on FLQA-250. This trend highlights the effectiveness of syntactic steering in reasoning scenarios when the target class is clear. It is moreover beneficial as noise mitigation, maintaining robust performance even in more challenging settings.

For ARI-ADD, we first want to highlight that all generated outputs, across all models, were numbers. The drop in accuracy therefore refers to generation of the wrong digits. Simply annotating the single digits with number metadata already improves accuracy by 4.3%. Through syntactic steering, i.e. prioring the model to continue with digits at the ␣ token of the prompt, proves crucial in unlocking the full potential of the model on the arithmetic task and improves accuracy by 36.1%.

## 5 DISCUSSION

Building on the promising results presented above, we now discuss key observations and potential directions for future work with `LIME` models.

**Pre-Training Efficiency.** `LIME` models consistently reach baseline token accuracy and perplexity with substantially fewer training tokens. This suggests that linguistic metadata provides meaningful information that default embeddings would otherwise need to learn expensively, and moreover do not converge to for 302 billion tokens. The observed benefits scale to larger model sizes; we conducted experiments with models of up to 2B parameters. `LIME` metadata embeddings reshape the inductive biases of LLMs. Whereas standard approaches compress linguistic and metadata signals into a single embedding space, `LIME` keeps them disentangled at the embedding mapping, offering weak supervision. Furthermore, we find that the language modeling improvements of `LIME` do not consistently generalize to standard downstream predictive tasks, i.p. those based on logit comparison, which is consistent with the observations of Tay et al. (2022) and Wettig et al. (2024). Providing the LLM with accurate look-ahead metadata, as in $\text{LIME}^{+1}$, acts as a constraint on the search space for the next token, leading to substantial improvements in accuracy and perplexity across model sizes.

**Scaling LIME Models.** The computational overhead of `LIME` models demonstrates robust scaling characteristics, as it increases linearly with the context length while remaining independent of all other architectural hyperparameters. Specifically, it runs on CPU at negligible costs compared to model execution, as described in App. A.3. When scaling model parameters, we observe a modest reduction in relative gains on training metrics such as perplexity as model size increases (see App. A.4). This may be due to the logarithmic nature of the scales. At the same time, metadata embeddings provide substantial benefits for larger models, including the $\text{LIME}_{2\text{B}}$ model, on tasks that involve emerging capabilities such as arithmetic (see App. A.5), even when trained solely on the base pre-training dataset without instruction tuning. These findings suggest that `LIME` scales effectively to larger models while still offering meaningful advantages in training data efficiency.

**Token Coupling.** Linguistic metadata helps models maintain coherence both within subword boundaries and beyond single-word tokens (Sec. 4.4). Our analyses show that `LIME` strengthens token coupling across words of all lengths and improves predictions for digits and entities. Using look-ahead metadata, as with $\text{LIME}^{+1}$, further amplifies accuracy gains. By binding fragmented tokens into coherent units, metadata embeddings not only boost accuracy but also resolve inconsistencies inherent to subword tokenization, such as handling prefixes, suffixes, and numbers, which even renders reasoning abilities more robust and precise. These results highlight metadata as a powerful inductive bias and a potential complement to tokenizer architectures, providing structural guidance that would otherwise be missing.

**Tokenizer's Language Bias.** Tokenization inherently introduces language-specific biases. Due to the fixed vocabulary size and the challenges of modifying embedding layers after training, underrepresented languages often undergo suboptimal segmentation. Such segmentations frequently misalign with semantic boundaries, which can negatively impact overall model performance. `Lime Tokenization` provides a simple yet effective way to extract a richer data signal per token by leveraging lightweight annotation models. As previously stated this has in particular been beneficial to keep split tokens together. Future work should explore augmenting the embedding space with explicit language identifiers to further improve multilingual robustness. Additionally, the tokenizer-agnostic nature of our method opens the door to extending it beyond subword-based tokenizers, including byte-level, character-level, or even tokenizer-free models (Deiseroth et al., 2024).

**Metadata Steering.** Metadata inference-time steering with $\text{LIME}^{+1}$ enables controllable generation. In our generative use cases (Sec. 4.5), metadata guidance leads to substantial improvements in reasoning and arithmetics. This indicates that metadata is not only a meaningful training signal but also a useful mechanism at inference, providing a new, interpretable interface for controllable token generation. Unlike fine-tuning methods or steering vector methods, metadata steering operates directly at the embedding layer, requiring no retraining or fine-tuning. The improvements of both `LIME` and $\text{LIME}^{+1}$ models on the addition task suggest that heterogeneity in the `Baseline` latent space hinders the early emergence of arithmetic capabilities. While this study focuses on linguistic metadata, our approach can be readily adapted to other domains. In some cases, non-linguistic domains may provide even more informative look-ahead metadata for language modeling. This should particularly be useful in the current research trend towards smaller agentic experts.

**Predicting Metadata.** Throughout this work we applied spaCy as a metadata annotator for POS and NER tags. Being a model itself, it achieves an accuracy of roughly 97% on POS and F-score of 86% on NER. Albeit not being perfect, we demonstrated consistently improved performance when applying these annotations during training. $\text{LIME}^{+1}$ leverages look-ahead metadata to achieve even stronger performance. Metadata steering, however, relies on knowledge of the next token. We demonstrated common use cases where this information is naturally available, allowing $\text{LIME}^{+1}$ to capitalize on these gains. However, in tasks where such information is absent, either additional supervision is required, or the model must learn to predict the appropriate next-token metadata itself. To show feasibility of the latter, we extended the language modeling head of a $\text{LIME}^{+1}_{2\text{B}}$ model with an additional metadata head tasked to predict look-ahead POS tags. After pre-training with a balanced loss on both heads, the metadata head achieves a top-3 accuracy of 82.32%, without affecting the language head's performance. Prior work has also explored prediction using intermediate internal representations of large language models (Popovic & Färber, 2024; Ghandeharioun et al., 2024). This demonstrates that simultaneous autoregressive prediction of look-ahead metadata holds strong potential, and, by further steering with it as shown with $\text{LIME}^{+1}$, leverages gains in token accuracy.

**Flexibility of Metadata.** Our method deliberately adopts a simple and transparent training configuration in which full input sequences are annotated at once without imposing constraints on how metadata is generated. The `LIME` method's design highlights the flexibility of this paradigm. Specifically, despite the absence of strict causality in our training experiments, when being evaluated under an enforced strictly causal regime (a condition outside its training distribution where failure is expected), the model exhibited robustness, maintaining performance parity with or demonstrating clear advantages over baseline models (see A.10). If strict causality of Metadata is desired, it can be incorporated through several straightforward strategies in model training. Subsets of tokens during training may be intentionally re-labeled or noised to improve robustness, and metadata predictors can be strengthened and jointly optimized as previously discussed. In practice, predictive metadata further reduces the relevance of strict causality by enabling models to infer reliable anticipatory signals on their own. It is also likely that `LIME`$^{+1}$ may be used in constrained generation or settings where next-token information is naturally available or by running auxiliary models alongside it. Finally, we observed that the spaCy models used may occasionally fail, e.g., on Q/A-style templates (see A.9). We left these errors uncorrected; addressing them would likely further improve the benchmark results. Overall, the design space for strictly causal or hybrid metadata-steered models is broad and supports multiple paths toward further improvement.

## 6 CONCLUSION

In this work, we introduced `LIME`, a novel method to overload token embeddings with linguistic metadata capturing syntax, semantics, and contextual information. Our approach demonstrates improvements in language modeling capabilities across various model sizes, while requiring minimal additional computational resources. We show that `LIME` models keep split word tokens together and improve cohesion of entities spanning tokens. Our method shows significant improvements on generative tasks. Furthermore, by pre-training with look-ahead metadata embeddings in `LIME`$^{+1}$ , we show that token generation can effectively be steered, which is particularly beneficial for noisy reasoning and arithmetic tasks. These results highlight that `LIME` is a seamless way of integrating metadata as an auxiliary data signal, enhancing both model efficiency and controllability. The gains observed with `LIME` and `LIME`$^{+1}$ demonstrates that models substantially benefit from access to lightweight metadata. As this benefit occurs even in models up to the 2-billion-parameter scale, it implies that linguistic features are not yet easily or exhaustively learned. This limitation suggests that performance gains could be considerably amplified if models were equipped with reliable predictors for upcoming token classes. Thus, our results emphasize the current efficacy of metadata steering while pointing to the substantial performance headroom achievable through the integration of richer anticipatory signals.

Future work could explore alternative methods for generating the look-ahead metadata used in `LIME`$^{+1}$, such as recovering it from the internal states of LLMs, to further enhance autoregressive capabilities. Additionally, evaluating `LIME` 's compatibility with different tokenization strategies and extending the approach beyond language modeling could unlock broader applications and capabilities. Further, the experiments in this work focused on English, as the `DCLM-Baseline` dataset is predominantly English. However, multilingual `LIME` models offer an interesting direction for future work. Several promising avenues include: (1) replacing language-specific POS embeddings with UPOS, a universal part-of-speech scheme with 17 labels designed for cross-linguistic consistency and applicable to many languages, which we expect to generalize well with appropriate taggers; (2) extending the POS layer to incorporate tag sets from additional languages; and (3) adding supplementary embedding layers, for instance, grouped by language or language family.

## 7 REPRODUCIBILITY

We are committed to ensuring the reproducibility of our work. Upon publication, we will release the code, the final pre-trained models, and detailed instructions necessary to reproduce all final experiments and results presented in this paper.

Meanwhile we have the modified python tokenizer class, the core of this research, attached. Furthermore, we report all hyperparameters in App. A.2.

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

## A   APPENDIX

The appendix includes training curves and numerical results from pre-training, along with the hyperparameters used for both the model architectures and training procedures. We further show the impact of `LIME` tokenization, and present extended benchmark results corresponding to Sec. 4.3. Sec. 4.4 is supplemented with comprehensive token accuracy tables, while Sec. 4.5 is supplemented with detailed FLenQA and ARI-ADD results together with examples of prompts and completions for each task. Finally, we provide the metadata annotator vocabularies employed throughout our experiments.

### A.1   LLM USAGE

We used LLMs to aid and polish writing our paper. The ideation, methodological design, and execution of experiments were solely the responsibility of the authors.

### A.2   LIME PRE-TRAINING SETUP DETAILS

Our 2B model architecture is derived from a Gemma-1 architecture with 2.43B parameters[5]. The 1B model architecture follows a Gemma-3-inspired design with 0.92B parameters[6]. Finally, the 500M architecture is based on a Gemma-3-style architecture with 0.50B trainable parameters. Tab. 4 presents a complete list of hyperparameter values. Exact parameter count can be found in Tab. 6. Training was conducted on 64 NVIDIA A100 GPUs (80GB each) across 8 nodes and required approximately 60 hours per model.

Table 4: Detailed list of hyperparameter values used for models and training.

| Category | Hyperparameters | 500M | 1B | 2B |
|---|---|---|---|---|
| Architecture | num_layers | 12 | 26 | 18 |
| | d_model | 768 | 1024 | 2048 |
| | mlp_factor | 6 | 6 | 8 |
| | num_heads | 4 | 4 | 8 |
| | num_kv_heads | 1 | 1 | 1 |
| | norm_type | RMS, $\epsilon = $ 1e-06 | | |
| | vocab_size | 256,000 | | |
| | position_embedding_type | rotary complex | | |
| | rotary_embedding_base | 10,000 | | |
| | activation_function | GELU | | |
| | mlp_bias | no | | |
| Optimization | loss_fn | Cross-Entropy | | |
| | optimizer | AdamW | | |
| | beta1, beta2, epsilon | [0.9, 0.95, 1.e-8] | | |
| | learning_rate | 3.e-3 | | |
| | lr_schedule | cosine decay to 3.e-4 | | |
| | warmup_steps | 3,600 (5%) | | |
| | weight_decay | 0.033 | | |
| | gradient_clipping | no | | |
| Stabilization | dropout | no | | |
| | attention_dropout | no | | |
| | embedding_dropout | no | | |
| | embedding_grad_scaling | inverse mini_batch freq. | | |
| | precision | bf16 | | |
| Training | global_batch_size | 2048 | 2048 | 512 |
| | sequence_length | 2048 | 2048 | 8192 |
| | micro_batch_size | 4 | 4 | 1 |
| | tokens_per_step | 4,194,304 | | |
| | steps | 72,000 | | |
| | packing_strategy | concatenation | | |

---

[5] https://huggingface.co/google/gemma-2b/blob/main/config.json
[6] https://huggingface.co/google/gemma-3-1b-pt/blob/main/config.json

## A.3 THE IMPACT OF LIME TOKENIZATION

We quantify the case where $n_{li} > n_{sw}$, meaning that $T_{li}$ produces more tokens from an input sequence than $T_{sw}$ (the Gemma tokenizer). By preserving the finer-grained pre-tokenization boundaries of $T_{li}$, Lime Tokenization may produce slightly more tokens. However, this tokenization maintains linguistically meaningful boundaries while reducing the effective vocabulary size by 1.03%. For instance, encoding 1,000 randomly selected DCLM-BASELINE samples (12.77M tokens) results in a 1.19% increase (12.90M tokens). Tab. 5 lists the most frequent sequences in this dataset where Lime Tokenization introduces additional granularity.

Table 5: Top 15 most frequent sequences where Lime Tokenization produces higher granularity when encoding 1,000 DCLM-BASELINE samples. Together, these sequences account for 54.18% of all cases with increased granularity.

| Gemma Tokenizer | Lime Tokenization | % |
|---|---|---|
| | | 8.67 |
| . | . | 6.53 |
| don | do n | 6.32 |
| ) . | ) . | 6.07 |
| ) , | ) , | 4.17 |
| . " | . " | 3.71 |
| . " | . " | 3.64 |
| didn | did n | 2.47 |
| doesn | does n | 2.45 |
| can | can | 2.29 |
| , " | , " | 1.92 |
| , " | , " | 1.73 |
| isn | is n | 1.59 |
| cannot | can not | 1.33 |
| . ) | . ) | 1.26 |

**Computational Cost.** The computational overhead of LIME Tokenization is dominated by the inference cost of the spaCy word-classification models we employ. These models are lightweight and run efficiently on CPUs, achieving throughput of up to 10,000 words per second [7]. Assuming pre-tokenization and annotation of 302B pre-training tokens and a conservative estimate of 1.5 tokens per word, this corresponds to roughly 5,583 CPU hours. Assuming preprocessing were performed prior to training, it would require approximately 10.9 hours of wall time on 8 nodes with 64 AMD EPYC 7F52 cores each, ahead of our 60-hour pre-training run. However, our training was not data-pipeline-bound and therefore we executed labeling on-the-fly and fully distributed the workload across data workers, resulting in virtually no additional pre-training wall time.

## A.4 LIME PRE-TRAINING RESULT DETAILS

Pre-training proceeded smoothly, with the loss decreasing consistently and gradient norms remaining stable. LIME$_{1B}$ required 44.64% fewer tokens to fit the training data distribution, while LIME$_{2B}$ required 34.76% fewer tokens (Fig. 5). As shown in Tab. 6 , LIME$^{+1}$ consistently improves accuracy by approximately 19.50% across all model sizes, with perplexity reductions more pronounced at 500M (13.95%) than at 2B (8.54%). As model size increases, the proportion of learnable parameters introduced by LIME decreases, reaching as little as 0.006% for the 2B model.

---

[7] https://spacy.io/usage/facts-figures#benchmarks-speed

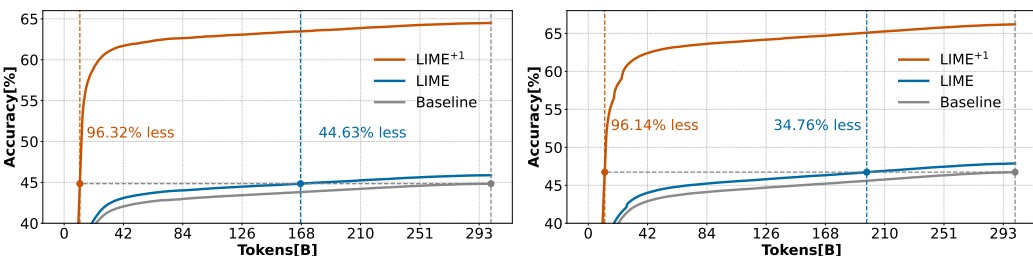

Figure 5: Next token prediction accuracy during Pre-training. **Left**: 1B models **Right**: 2B models.

Table 6: Overview of pre-training metrics and parameter scaling across model sizes.

| Size | Model | Token Accuracy | Perplexity | Total Parameters | LIME Parameters |
|------|-------|----------------|------------|------------------|-----------------|
| 500M | Baseline | 41.90 | 19.72 | 495,864,576 | |
| | LIME | 42.91 ↑ 1.01 | 18.50 ↓ 1.22 | 495,919,104 | +54,528 +0,0110% |
| | LIME$^{+1}$ | 61.83 ↑ 19.93 | 5.77 ↓ 13.95 | 495,919,104 | +54,528 +0,0110% |
| 1B | Baseline | 45.20 | 14.84 | 919,655,424 | |
| | LIME | 46.21 ↑ 1.01 | 14.02 ↓ 0.82 | 919,728,128 | +72,704 +0,0079% |
| | LIME$^{+1}$ | 64.86 ↑ 19.65 | 4.73 ↓ 10.11 | 919,728,128 | +72,704 +0,0079% |
| 2B | Baseline | 46.92 | 12.82 | 2,426,480,640 | |
| | LIME | 47.95 ↑ 1.03 | 12.10 ↓ 0.72 | 2,426,626,048 | +145,408 +0,0060% |
| | LIME$^{+1}$ | 66.40 ↑ 19.48 | 4.28 ↓ 8.54 | 2,426,626,048 | +145,408 +0,0060% |

## A.5 DETAILED BENCHMARK RESULTS

Benchmark results across all model sizes are listed in Tab. 7. All tasks are evaluated using a randomly selected 5-shot context on 10,000 samples, with standard errors reported.

Table 7: Detailed benchmark results across all model sizes. Improvements relative to the respective Baseline are indicated by ↑, relative to LIME with ⇑. Generative-format tasks are highlighted.

| | 500M | | | 1B | | | 2B | | |
|---|------|------|------|------|------|------|------|------|------|
| | Base | LIME | LIME$^{+1}$ | Base | LIME | LIME$^{+1}$ | Base | LIME | LIME$^{+1}$ |
| ARC-Easy [7] | 57.00±1.57 ↑ | 57.20±1.57 | ⇑58.10±1.56 | 67.30±1.48 ↑ | 68.40±1.47 | ⇑68.80±1.47 | 71.20±1.43 ↑ | 73.40±1.40 ↑ | 72.50±1.41 |
| BoolQ [6] | 49.50±1.58 | 47.90±1.58 | ⇑59.40±1.55 | 51.60±1.58 | 51.60±1.58 | ⇑60.70±1.55 | 64.10±1.52 | 63.10±1.53 | 60.30±1.55 |
| COPA [35] | 62.00±4.88 ↑ | 64.00±4.82 | 61.00±4.90 | 72.00±4.51 | 67.00±4.73 | 71.00±4.56 | 77.00±4.23 ↑ | 81.00±3.94 | ⇑83.00±3.78 |
| HellaSwag [51] | 36.20±1.52 ↑ | 37.00±1.53 | ⇑43.10±1.57 | 52.90±1.58 ↑ | 53.20±1.58 | ⇑55.60±1.57 | 63.30±1.52 | 61.50±1.54 | ⇑64.40±1.51 |
| LAMBADA [30] | 26.50±1.40 ↑ | 29.80±1.45 | ⇑49.00±1.58 | 41.00±1.56 ↑ | 46.00±1.58 | ⇑66.00±1.50 | 48.00±1.58 ↑ | 49.50±1.58 | ⇑73.40±1.40 |
| OpenBookQA [26] | 31.00±2.07 ↑ | 32.60±2.10 | ⇑34.20±2.12 | 36.40±2.15 ↑ | 37.80±2.17 | ⇑39.20±2.19 | 39.80±2.19 ↑ | 42.00±2.21 | ⇑43.00±2.22 |
| PIQA [4] | 69.90±1.45 | 69.40±1.46 | 68.20±1.47 | 72.60±1.41 ↑ | 73.20±1.40 | 71.20±1.43 | 74.10±1.39 ↑ | 75.10±1.37 | 73.70±1.39 |
| TriviaQA [16] | 8.20±0.87 ↑ | 9.80±0.94 | ⇑19.50±1.25 | 17.90±1.21 ↑ | 19.70±1.26 | ⇑33.90±1.50 | 25.80±1.38 | 25.00±1.37 | ⇑42.20±1.56 |
| WinoGrande [36] | 51.70±1.58 ↑ | 52.60±1.58 | ⇑54.60±1.58 | 57.40±1.56 ↑ | 57.60±1.56 | ⇑59.20±1.55 | 64.20±1.52 | 62.60±1.53 | 62.00±1.54 |
| Mean | 43.56±1.88 ↑ | 44.48±1.89 | ⇑49.68±1.95 | 52.12±1.90 ↑ | 52.71±1.90 | ⇑58.40±1.92 | 53.52±2.19 ↑ | 53.95±2.18 | ⇑63.83±1.82 |

## A.6 TOKEN COUPLING DETAILS

Fig. 6 illustrates that summing all positive and negative token prediction accuracy changes from our qualitative analysis in Sec. 4.4 reproduces the overall LIME$_{500M}$ accuracy improvement of 1.01% reported in Sec. 4.2 .

In our experiments, we define natural language words as sequences of $x$ alphabetic character tokens ([A-Za-z]) enclosed by whitespaces. Additionally, we include hyphenated compound words, e.g. *long-term* and words bound by the apostrophe ' covering possessions and contractions, e.g., *Murphy's* and *We'll*. Tab. 8 provides a detailed view of how token prediction accuracy varies for words of up to $x = 6$ tokens. Additionally, Tab. 9 presents accuracies calculated in the same way but *includes* the first token of each word. While the absolute values and improvements are smaller, the trend across word lengths remains consistent, and no outliers are observed. This suggests that the observed coupling improvements are not notably influenced by the model correctly predicting the first token of a word.

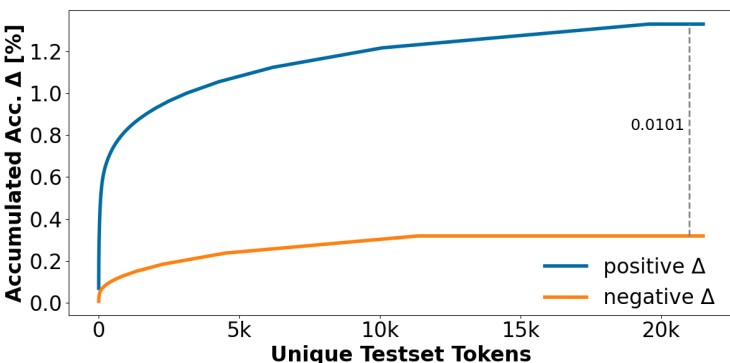

Figure 6: Accuracy shifts resemble the initially observed accuracy improvement of our qualitative analysis of LIME$_{500M}$ and Baseline.

Table 8: Token accuracy within words of length $x$ improves across all model sizes.

| Model Size | $x$ | $n$ | Baseline | LIME | LIME$^{+1}$ |
|---|---|---|---|---|---|
| 500M | 2 | 164,256 | 69.63 | 76.34 ↑ 6.71 | 79.80 ↑ 10.17 |
| 500M | 3 | 174,852 | 68.48 | 71.81 ↑ 3.33 | 80.84 ↑ 12.36 |
| 500M | 4 | 61,947 | 75.36 | 76.94 ↑ 1.58 | 82.09 ↑ 6.73 |
| 500M | 5 | 16,152 | 66.98 | 68.57 ↑ 1.59 | 80.18 ↑ 13.20 |
| 500M | 6 | 2,840 | 55.95 | 56.65 ↑ 0.70 | 67.82 ↑ 11.87 |
| 1B | 2 | 164,256 | 73.94 | 80.23 ↑ 6.29 | 83.31 ↑ 9.37 |
| 1B | 3 | 174,852 | 71.87 | 75.13 ↑ 3.26 | 83.49 ↑ 11.62 |
| 1B | 4 | 61,947 | 77.88 | 79.61 ↑ 1.73 | 84.68 ↑ 6.80 |
| 1B | 5 | 16,152 | 70.38 | 72.91 ↑ 2.53 | 83.36 ↑ 12.98 |
| 1B | 6 | 2,840 | 60.39 | 63.24 ↑ 2.85 | 71.97 ↑ 11.58 |
| 2B | 2 | 164,256 | 76.46 | 82.24 ↑ 5.78 | 85.06 ↑ 8.60 |
| 2B | 3 | 174,852 | 73.77 | 76.78 ↑ 3.01 | 84.72 ↑ 10.95 |
| 2B | 4 | 61,947 | 79.92 | 81.00 ↑ 1.08 | 86.34 ↑ 6.42 |
| 2B | 5 | 16,152 | 72.46 | 74.41 ↑ 1.95 | 84.73 ↑ 12.27 |
| 2B | 6 | 2,840 | 64.61 | 66.20 ↑ 1.59 | 74.82 ↑ 10.21 |
| 500M | ≥2 | 424,960 | 69.68 | 73.95 ↑ 4.27 | 80.41 ↑ 10.73 |
| 1B | ≥2 | 424,960 | 73.32 | 77.48 ↑ 4.16 | 83.43 ↑ 10.11 |
| 2B | ≥2 | 424,960 | 75.51 | 79.24 ↑ 3.73 | 84.95 ↑ 9.44 |
| 500M | 1 | 5,049,553 | 33.91 | 34.59 ↑ 0.68 | 55.10 ↑ 21.19 |
| 1B | 1 | 5,049,553 | 37.23 | 37.89 ↑ 0.66 | 58.35 ↑ 21.12 |
| 2B | 1 | 5,049,553 | 39.08 | 39.80 ↑ 0.72 | 59.94 ↑ 20.86 |

Table 9: The trend of token-level accuracy improvements for words of length $x$ persists even when the first token of each word is included.

| Model Size | $x$ | $n$ | Baseline | LIME | LIME$^{+1}$ |
|---|---|---|---|---|---|
| 500M | 1 | 5,049,553 | 33.91 | 34.59 ↑ 0.68 | 55.10 ↑ 21.19 |
| 500M | 2 | 328,512 | 43.28 | 46.86 ↑ 3.58 | 54.15 ↑ 10.87 |
| 500M | 3 | 262,278 | 51.89 | 54.29 ↑ 2.40 | 66.23 ↑ 14.34 |
| 500M | 4 | 82,596 | 60.97 | 62.32 ↑ 1.35 | 69.80 ↑ 8.83 |
| 500M | 5 | 20,190 | 55.88 | 57.29 ↑ 1.41 | 69.36 ↑ 13.48 |
| 500M | 6 | 3,408 | 49.32 | 50.15 ↑ 0.83 | 61.30 ↑ 11.98 |
| 1B | 1 | 5,049,553 | 37.23 | 37.89 ↑ 0.66 | 58.35 ↑ 21.12 |
| 1B | 2 | 328,512 | 47.35 | 50.79 ↑ 3.44 | 58.57 ↑ 11.22 |
| 1B | 3 | 262,278 | 55.27 | 57.66 ↑ 2.39 | 69.67 ↑ 14.40 |
| 1B | 4 | 82,596 | 63.67 | 65.14 ↑ 1.47 | 72.73 ↑ 9.06 |
| 1B | 5 | 20,190 | 59.26 | 61.37 ↑ 2.11 | 72.97 ↑ 13.71 |
| 1B | 6 | 3,408 | 53.58 | 55.96 ↑ 2.38 | 65.02 ↑ 11.44 |
| 2B | 1 | 5,049,553 | 39.08 | 39.80 ↑ 0.72 | 59.94 ↑ 20.86 |
| 2B | 2 | 328,512 | 49.79 | 53.00 ↑ 3.21 | 60.87 ↑ 11.08 |
| 2B | 3 | 262,278 | 57.25 | 59.42 ↑ 2.17 | 71.21 ↑ 13.96 |
| 2B | 4 | 82,596 | 65.68 | 66.63 ↑ 0.95 | 74.59 ↑ 8.91 |
| 2B | 5 | 20,190 | 61.26 | 62.98 ↑ 1.72 | 74.57 ↑ 13.31 |
| 2B | 6 | 3,408 | 57.28 | 58.74 ↑ 1.46 | 68.10 ↑ 10.83 |

## A.7 EXTENSIVE FLENQA AND ARI-ADD RESULTS

Here, we present the complete results for reasoning and arithmetic tasks across all model sizes. Tab. 10 shows that the evaluated arithmetic capabilities begin to emerge at the 1B scale, with LIME providing slight improvements and LIME$^{+1}$ yielding a substantial increase of 12.9%.

Table 10: Complete ARI-ADD results.

| Size | Baseline | LIME | LIME$^{+1}$ |
|------|---------|------|------------|
| 2B   | 22.6    | 26.9 ↑ 4.3 | 58.7 ↑ 36.1 |
| 1B   | 4.1     | 5.2 ↑ 1.1  | 17.0 ↑ 12.9 |
| 500M | 0.5     | 0.1 ↓ 0.4  | 1.3 ↑ 0.8 |

Tab. 11 reports FLenQA results for all model sizes and applicable noise lengths. Variations in spaces and dashes between two words are ignored when matching the ground truth to avoid penalizing correctly predicted but differently formatted completions. LIME$^{+1}$ consistently improves reasoning performance across scales, with the 500M model showing the largest gain of 51.8% on FLQA-500.

Table 11: Detailed FLenQA results across model sizes. While Baseline and LIME match the first eight generated words, LIME$^{+1}$ generates and matches only one word.

| Size | Task | Baseline | LIME | LIME$^{+1}$ |
|------|------|---------|------|------------|
| 2B   | FLQA-250  | 42.0 | 52.0 ↑ 10.0 | 80.0 ↑ 38.0 |
| 2B   | FLQA-500  | 49.5 | 65.3 ↑ 15.8 | 73.5 ↑ 24.0 |
| 2B   | FLQA-1000 | 34.8 | 47.0 ↑ 12.2 | 65.3 ↑ 30.5 |
| 2B   | FLQA-2000 | 40.3 | 44.3 ↑ 4.0  | 52.5 ↑ 12.2 |
| 2B   | FLQA-3000 | 28.2 | 30.0 ↑ 1.8  | 39.3 ↑ 11.1 |
| 1B   | FLQA-250  | 36.0 | 28.0 ↓ 8.0  | 80.0 ↑ 44.0 |
| 1B   | FLQA-500  | 39.0 | 32.0 ↓ 7.0  | 74.0 ↑ 35.0 |
| 1B   | FLQA-1000 | 32.4 | 33.0 ↑ 0.6  | 78.5 ↑ 46.1 |
| 500M | FLQA-250  | 22.0 | 40.0 ↑ 18.0 | 70.0 ↑ 48.0 |
| 500M | FLQA-500  | 22.0 | 48.8 ↑ 26.8 | 73.8 ↑ 51.8 |
| 500M | FLQA-1000 | 12.5 | 30.5 ↑ 18.0 | 72.5 ↑ 42.0 |

Additionally we present FLenQA results with a different prompt and stricter matching (Tab. 12). The prompt is shorter and less specific (*Ethan is in a*), and the matching criteria are more stringent (first-two words). Under these conditions, LIME$_{1B}$ shows substantial improvement, while Baseline and LIME perform generally lower. LIME$^{+1}$ improvements across model sizes are also amplified, reaching +75.8% for LIME$^{+1}_{500M}$.

Table 12: Additional FLenQA results evaluated with strict next-two-words matching using the prompt *'is in a'* .

| Size | Task | Baseline | LIME | LIME$^{+1}$ |
|------|------|---------|------|------------|
| 2B   | FLQA-250  | 12.0 | 22.0 ↑ 10.0 | 76.0 ↑ 64.0 |
| 2B   | FLQA-500  | 12.5 | 15.3 ↑ 2.8  | 62.3 ↑ 49.8 |
| 2B   | FLQA-1000 | 7.3  | 9.5 ↑ 2.2   | 63.8 ↑ 56.5 |
| 2B   | FLQA-2000 | 5.0  | 7.0 ↑ 2.0   | 55.8 ↑ 50.8 |
| 2B   | FLQA-3000 | 3.8  | 7.8 ↑ 4.0   | 52.5 ↑ 48.7 |
| 1B   | FLQA-250  | 32.0 | 58.0 ↑ 26.0 | 98.0 ↑ 66.0 |
| 1B   | FLQA-500  | 6.0  | 32.5 ↑ 26.5 | 78.0 ↑ 72.0 |
| 1B   | FLQA-1000 | 4.3  | 24.8 ↑ 20.5 | 68.8 ↑ 64.5 |
| 500M | FLQA-250  | 10.0 | 14.0 ↑ 4.0  | 80.0 ↑ 70.0 |
| 500M | FLQA-500  | 3.5  | 4.0 ↑ 0.5   | 79.3 ↑ 75.8 |
| 500M | FLQA-1000 | 2.3  | 1.5 ↓ 0.8   | 68.8 ↑ 66.5 |

## A.8 ARI-ADD AND FLENQA COMPLETIONS

We illustrate FLenQA and ARI-ADD evaluations using example prompts and completions from the 2B models (Tab. 13). In the two sampled tokens shown, the completions consist solely of digits, a pattern that holds for all completions across all model variants. This demonstrates that LIME and LIME$^{+1}$ enhance arithmetic capabilities rather than just predicting digits.

Table 13: ARI-ADD prompt and completion examples generated by our 2B models.

| Prompt | Completion | Model |
|---|---|---|
| The result is: 12+15 = | 27 | Baseline |
| | 27 | LIME |
| | 27 | LIME$^{+1}$ |
| The result is: 18+26 = | 36 | Baseline |
| | 44 | LIME |
| | 44 | LIME$^{+1}$ |
| The result is: 48+45 = | 10 | Baseline |
| | 91 | LIME |
| | 93 | LIME$^{+1}$ |

Further, Tab. 14 contains prompt and completion examples from our FLenQA evaluation, specifically from the 2B model with 1,000 noise (FLQA-1000). Ground truth is matched within the first eight words, while LIME$^{+1}$ uses only one. For comparability, words such as *white walled* are counted as a single word, consistent with counting *marble-floored* as one word.

Table 14: FLQA-1000 prompt and completion examples generated by our 2B models.

| Prompt | Completion | Model |
|---|---|---|
| Eric George is in a room. It has the following properties: | \n • It is a **white walled**. \n • It is | Baseline |
| | it is **white walled**, it is a room, and | LIME |
| | **white walled** | LIME$^{+1}$ |
| Kathleen Russo is in a room. It has the following properties: | \n - it is a room \n - it is a room | Baseline |
| | It is **marble-floored**.\n It is a foyer. It | LIME |
| | **marble-floored** | LIME$^{+1}$ |
| Rachel Hancock is in a room. It has the following properties: | it is a room, it is a room | Baseline |
| | it is a room, it is a room | LIME |
| | **wooden-floored** | LIME$^{+1}$ |

## A.9 Noise Robustness of LIME models

We investigate the role of label noise further by introducing controlled POS and NER noise levels of $0.001 \leq p \leq 0.3$ during teacher-forced next-token accuracy evaluation. The spaCy-predicted label was replaced with a randomly selected label with probability $p$. Note that this form of noise is out-of-distribution, since our models were trained solely on spaCy-generated labels without any artificial label noise. We observe that $LIME_{500M}$ and $LIME_{2B}$ exhibit comparable robustness under increasing noise, whereas the $LIME_{1B}$ model degrades more rapidly, an observation we consider particularly noteworthy (see App. 7). We assume that this may relate to skewed architectural differences, specifically the larger number of transformer blocks (see App. A.2) in the 1B model compared to the 500M and 2B models. Further, we want to emphasize, that the model has already been trained with imperfect labels, exposing it to some noise. While the overall noise was small, adding artificial noise at inference quickly pushes token predictions out of distribution. This behavior is not only common for POS or NER labels but also for standard LLM token prediction. Large noise levels (over a few percent) are unrealistic and shown only for illustration; smaller amounts are more typical, and within this range the model remains relatively robust. Given the imperfection of annotation models, improving them could further enhance both token-level accuracy and downstream performance, benefiting LIME as well.

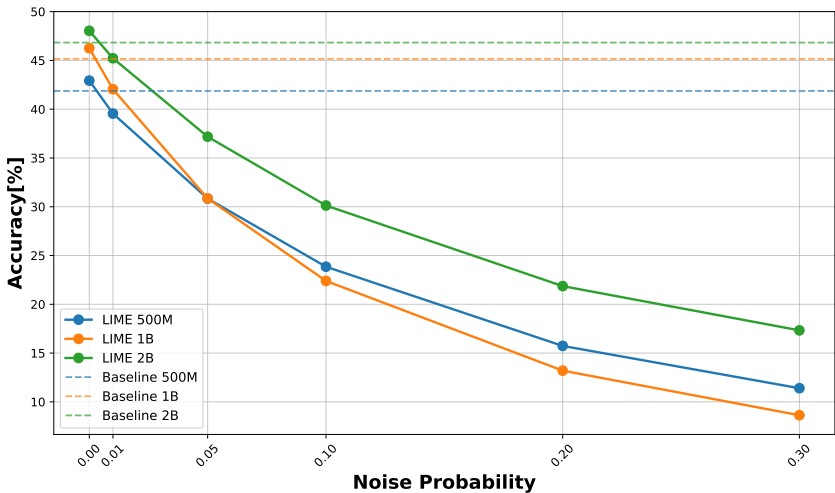

Figure 7: LIME next token prediction accuracy under increasingly noisy metadata labels.

Table 15: Accuracy (%) under artificial noise levels for different model sizes.

| $p$ | 0 | 0.01 | 0.05 | 0.1 | 0.2 | 0.3 |
|---|---|---|---|---|---|---|
| LIME 500M | 42.93 | 39.55 | 30.84 | 23.85 | 15.73 | 11.40 |
| LIME 1B | 46.25 | 42.04 | 30.85 | 22.39 | 13.20 | 8.63 |
| LIME 2B | 48.03 | 45.22 | 37.18 | 30.13 | 21.86 | 17.33 |

## A.10 Strictly causal Evaluation

Bidirectional metadata annotators such as spaCy's POS and NER models can assign labels that depend on surrounding tokens, which may introduce inconsistencies in a strictly causal, one-by-one token generation setting. Thus, we investigated the impact of causal information in metadata annotations on LIME models. To this end, we re-evaluated training metrics (Sec. 4.2) and downstream benchmarks (Sec. 4.3), ensuring that each token's metadata was not influenced by future tokens. First, we observe that LIME models remain on par with the baseline, even in this out-of-training-distribution setting, although the improvements are marginally reduced (see Tab. 16), demonstrating strong robustness and adaptability. Second, downstream performance is similarly unaffected (see Tab. 17). Finally, in the Q/A setting, we observed noise in spaCy-generated metadata; for example, Edmund is labeled

as an Adjective when preceded by `Answer:` but as a Noun otherwise. This highlights that `LIME` maintains strong performance even under suboptimal labeling conditions. Importantly, our work does not focus on metadata retrieval but instead establishes valid upper bounds for performance when such data is available, particularly highlighted by our $LIME^{+1}$ models.

Table 16: Next token prediction accuracy in a strictly causal setting.

| | 500M | | 1B | | 2B | |
|---|---|---|---|---|---|---|
| | Baseline | LIME | Baseline | LIME | Baseline | LIME |
| Accuracy [%] | 41.458 | 41.640 | 44.756 | 44.803 | 46.379 | 46.430 |

Table 17: Benchmark results in a strictly causal, one-by-one token generation setting.

| | 500M | | 1B | | 2B | |
|---|---|---|---|---|---|---|
| | Baseline | LIME | Baseline | LIME | Baseline | LIME |
| ARC-Easy | 56.80 ±1.57 | 56.10 ±1.57 | 67.50 ±1.48 | 66.70 ±1.49 | 71.30 ±1.43 | 70.50 ±1.44 |
| BoolQ | 49.50 ±1.58 | 47.90 ±1.58 | 51.60 ±1.58 | 51.60 ±1.58 | 64.10 ±1.52 | 63.20 ±1.53 |
| COPA | 62.00 ±4.88 | 66.00 ±4.76 | 72.00 ±4.51 | 71.00 ±4.56 | 77.00 ±4.23 | 77.00 ±4.23 |
| HellaSwag | 36.20 ±1.52 | 37.50 ±1.53 | 53.40 ±1.58 | 52.00 ±1.58 | 63.80 ±1.52 | 58.00 ±1.56 |
| LAMBADA | 26.50 ±1.40 | 28.70 ±1.43 | 41.00 ±1.56 | 42.70 ±1.56 | 48.00 ±1.58 | 46.10 ±1.58 |
| OpenBookQA | 31.00 ±2.07 | 29.20 ±2.04 | 36.60 ±2.16 | 34.00 ±2.12 | 39.80 ±2.19 | 38.00 ±2.17 |
| PIQA | 69.50 ±1.46 | 68.10 ±1.47 | 72.70 ±1.41 | 70.40 ±1.44 | 74.60 ±1.38 | 72.80 ±1.41 |
| TriviaQA | 8.20 ±0.87 | 7.90 ±0.85 | 17.90 ±1.21 | 16.20 ±1.17 | 25.80 ±1.38 | 20.10 ±1.27 |
| WinoGrande | 50.90 ±1.58 | 50.90 ±1.58 | 57.50 ±1.56 | 56.20 ±1.57 | 64.20 ±1.52 | 60.80 ±1.54 |
| Mean | 43.40 ±1.88 | 43.59 ±1.87 | 52.24 ±1.89 | 51.20 ±1.90 | 58.73 ±1.86 | 56.28 ±1.86 |

## A.11 METADATA ANNOTATOR VOCABULARY: POS (ENGLISH)

For POS metadata embeddings, we use the spaCy `en_core_web_sm` model containing a lightweight CPU-based POS tagger that returns 50 tags (Tab. 18). `Lime Tokenization` augments its vocabulary with one additional special token.

Table 18: Glossary of 51 POS tags with their respective descriptions.

| | |
|---|---|
| . | punctuation mark, sentence closer |
| , | punctuation mark, comma |
| -LRB- | left round bracket |
| -RRB- | right round bracket |
| `` | opening quotation mark |
| '' | closing quotation mark |
| : | punctuation mark, colon or ellipsis |
| $ | symbol, currency |
| AFX | affix |
| CC | conjunction, coordinating |
| CD | cardinal number |
| DT | determiner |
| EX | existential there |
| FW | foreign word |
| HYPH | punctuation mark, hyphen |
| IN | conjunction, subordinating or preposition |
| JJ | adjective (English), other noun-modifier (Chinese) |
| JJR | adjective, comparative |
| JJS | adjective, superlative |
| LS | list item marker |
| MD | verb, modal auxiliary |
| NN | noun, singular or mass |
| NNP | noun, proper singular |
| NNPS | noun, proper plural |
| NNS | noun, plural |
| PDT | predeterminer |
| POS | possessive ending |
| PRP | pronoun, personal |
| PRP$ | pronoun, possessive |
| RB | adverb |
| RBR | adverb, comparative |
| RBS | adverb, superlative |
| RP | adverb, particle |
| TO | infinitival "to" |
| UH | interjection |
| VB | verb, base form |
| VBD | verb, past tense |
| VBG | verb, gerund or present participle |
| VBN | verb, past participle |
| VBP | verb, non-3rd person singular present |
| VBZ | verb, 3rd person singular present |
| WDT | wh-determiner |
| WP | wh-pronoun, personal |
| WP$ | wh-pronoun, possessive |
| WRB | wh-adverb |
| SP | space (English), sentence-final particle (Chinese) |
| ADD | email |
| NFP | superfluous punctuation |
| XX | unknown |
| _SP | whitespace |
| SPECIAL | special token |

## A.12 METADATA ANNOTATOR VOCABULARY: NER

For NER metadata embeddings, we employ the spaCy `en_core_web_sm` model, using its lightweight CPU-based NER tagger. The tagger outputs 20 tags (Tab. 19) and `LIME Tokenization` extends its vocabulary with one additional special token.

Table 19: Glossary of 20 NER tags with their respective descriptions.

| | |
|---|---|
| PERSON | People, including fictional |
| NORP | Nationalities or religious or political groups |
| FAC | Buildings, airports, highways, bridges, etc. |
| ORG | Companies, agencies, institutions, etc. |
| GPE | Countries, cities, states |
| LOC | Non-GPE locations, mountain ranges, bodies of water |
| PRODUCT | Objects, vehicles, foods, etc. (not services) |
| EVENT | Named hurricanes, battles, wars, sports events, etc. |
| WORK_OF_ART | Titles of books, songs, etc. |
| LAW | Named documents made into laws |
| LANGUAGE | Any named language |
| DATE | Absolute or relative dates or periods |
| TIME | Times smaller than a day |
| PERCENT | Percentage, including "%" |
| MONEY | Monetary values, including unit |
| QUANTITY | Measurements, as of weight or distance |
| ORDINAL | "first", "second", etc. |
| CARDINAL | Numerals that do not fall under another type |
| " | No entity |
| SPECIAL | special token |

