# OpenReview forum: "$\texttt{LIME}$: Making LLM Data More Efficient with Linguistic Metadata Embeddings"
_ICLR.cc/2026/Conference — Submitted to ICLR 2026_

### Official Review · Reviewer_rBYA · 2025-10-23

**Soundness:** 3
**Presentation:** 2
**Contribution:** 2
**Rating:** 4
**Confidence:** 3

**Summary:**

This paper proposes LIME and LIME+1, methods for incorporating token-level linguistic metadata (POS/NER tags) into the embedding layer of large language models. While the idea is simple and yields modest improvements in data efficiency, the LIME+1 variant raises significant conceptual and practical concerns.

**Strengths:**

The proposal is minimal architectural modification (just adding metadata embeddings). Implementation is straightforward, and the overhead in parameters/compute is negligible.

The data-efficiency gains reported are interesting (e.g., same loss with ~56% less data at 500M).

Analysis of subword cohesion and token coupling offers some interpretability. In fact, the approach in LIME+1 is indeed interesting and promising since it also allows faster adaptation during pretraining stage.

**Weaknesses:**

1) (Minor Weakness) The core idea of injecting linguistic or syntactic tags into embeddings has been explored in earlier works. For example, see [1,2].

2) Experiments are limited to English web data. However, metadata extraction quality degrades on noisy or multilingual corpora. It’s unknown whether LIME works with other data, languages.

3) While the direction is interesting, the LIME+1 results rely on handcrafted oracle metadata and thus overstate the method’s practical impact. The causal inconsistency substantially weakens the contribution.

For now I recommend rejection, but I'm open to increase the score if the authors can provide a learnable or inference-consistent variant of LIME+1.

[1] Armengol-Estape, Jordi, Marta R. Costa-jussà, and Carlos Escolano. "Enriching the Transformer with Linguistic Factors for Low-Resource Machine Translation." Proceedings of the International Conference on Recent Advances in Natural Language Processing (RANLP 2021). 2021.

[2] Sennrich, Rico, and Barry Haddow. "Linguistic Input Features Improve Neural Machine Translation." Proceedings of the First Conference on Machine Translation: Volume 1, Research Papers. 2016.

**Questions:**

1) Did you try LIME on data with spelling error?

2) How should one apply the LIME on other languages?

3) I do not clearly understand how POS/NER tags are generated during inference in LIME+, can you further explain it?

---

> ### Author Response · Authors · 2025-11-20
>
> Dear Reviewer **rBYA**,
>
> thank you for the constructive feedback. We respond to the raised points below.
>
> ---
>
> ### *Q1. Spelling errors*
>
> In the context of LIME, spelling errors have two effects: (a) they change the tokenization, and (b) they can invalidate metadata. In the case of (a) alone, with correct metadata, the metadata could help correct the effect. Similar to LLMs’ robustness to spelling errors, the model would need to be trained on such errors to handle them effectively. While we do not test extending spaCy, we do train a metadata prediction head from scratch, suggesting that achieving spelling error robustness is feasible.
>
> Therefore, we did not explicitly train LIME on spelling errors, though this represents an interesting extension for future work. Instead, we address the concern of noisy data through targeted experiments that evaluate the effect of noisy labels at inference time by repeating our teacher-forced next-token accuracy evaluation of Section 4.2. We introduce progressively increasing noise, where the spaCy-predicted label was replaced with a randomly selected label with probability `p=0.01` to `0.3`. Note, that this form falls out of training-data distribution, since our models were trained solely on spaCy-generated labels without any artificial label noise.
>
> As shown in the attached Figure [1], next-token accuracy decreases smoothly for the 500M and 2B models, while the 1B model degrades more rapidly. We find this divergence particularly interesting and hypothesize that it may stem from architectural differences, specifically, the higher number of transformer blocks in the 1B model (see Appendix Table 4).
>
> ---
>
> ### *Q2 + W2. Multilingual inputs*
>
> This is an interesting question and was likewise noted by reviewer **LK2J**.
> In principle, both LIME and LIME+1 can be applied to any language and any type of metadata, provided suitable annotators (e.g., NLP models or other labeling tools) are available. There is, for instance, a broad list of spaCy models available [2], though this might require language classification upfront. Our experiments focused on English because the DCLM-baseline dataset is predominantly English, but multilingual LIME models represent an interesting direction for future work. We see several promising avenues to explore: (1) replacing language-specific POS embeddings with UPOS, a universal part-of-speech scheme with 17 labels designed for cross-linguistic consistency and applicable to many languages, which we expect to generalize effectively given suitable taggers; (2) expanding the POS layer to include tag sets from additional languages; and (3) introducing supplementary embedding layers, for example grouped by language or language family.
>
> ---
>
> ### *Q3. LIME+1 tag generation*
>
> LIME+1 models are trained using shifted metadata embeddings, and at inference time the model likewise expects the metadata label of the token it is predicting, we therefore call it metadata-steering. Although future metadata labels are fully available during training, the metadata classes from the final token of the prompt onward are manually specified to steer generation.
> For example, in the ARI-ADD prompt sequence from Section 4.5, all prompt tokens are provided with the metadata class of their successor token and the final prompt token, "␣", which is assigned the metadata label corresponding to a digit. We provide initial evidence that these future labels are learnable by pre-training a model with an additional POS annotation head for predicting future metadata, which can be straightforward in certain cases or for tokens where the next token is highly predictable.
>
> ---

---

> ### Author Response · Authors · 2025-11-20
>
> Additionally, we would like to address the weaknesses reviewer rBYA highlighted:
>
> ---
>
> ### *W1. Metadata embeddings*
>
> We appreciate the reviewer’s observation that incorporating linguistic metadata at the embedding level has been previously explored. Our work introduces several key differences: (1) we apply metadata embeddings beyond translation tasks to causal language modeling using decoder-only models at a considerably larger scale, (2) LIME is not restricted to part-of-speech metadata, and (3) we study shifted metadata embeddings and demonstrate their role in enabling metadata steering through the LIME+1 variant. We will expand the related work section to include the interesting work of Armengol-Estapé et al.
>
> ---
>
> ### *W3. LIME+1 variant*
>
> We would like to clarify the scope of LIME+1. As noted in our paper, LIME+1 is a task-specific approach, intended for use cases where it provides clear value (see further discussion below). In addition, it serves as a fundamentally important ablation study to better understand the effects of overloading embeddings. Therefore, we disagree that the method is unrealistic. At the same time, we acknowledge that it is not a general-purpose approach, a point we have already mentioned in the paper but will emphasize now more prominently in the revised version.
>
> In particular, LIME+1 serves as a foundational study, examining how knowledge of the next token's metadata can influence not only pre-training metrics but also performance on downstream tasks. Although this represents an idealized scenario, our results, especially those in Section 4.5, highlight both the promising potential of this approach and its relevance beyond a purely theoretical setting. We are encouraged by three observations: 1) Inferring POS/NER tags from hidden states is an active area of research [3][4], and we provide initial evidence by successfully training a model with an additional annotation prediction head. 2) Small multi-agent expert predictors are gaining popularity, making a dedicated POS/NER predictor in tandem with the main model an appealing strategy. 3) Our use cases demonstrate concrete scenarios with available future metadata where LIME+1 delivers substantial gains, despite being trained on the same data.
>
> Further, we would like to emphasize that all models (Baseline, LIME, LIME+1) are trained on the same data, with the only difference being the inclusion of metadata. This allows the results to clearly show the performance gains that access to this metadata can provide.
>
> Finally note that our LIME models indeed work in a strictly autoregressive setting. We measured around 9% of label-changes in strictly-causal corrections, while the error rate of spaCy is claimed to be 3-14% depending on the category. Even not being explicitly being trained for it, we successfully ran the models strictly causal--without any special adaptions, accepting false tags on incomplete tokens. We measured a drop in the accuracy of the next-token prediction of Section 4.2 from 0.8 to 0.3% for LIME `500M`. This drop should further be augmented by more explicit training and sampling strategies, e.g. through error tags. We will finalize the strictly auto-regressive results for all evaluations and add them to the appendix.
>
> ---
>
> We once again thank the reviewer for their time and constructive feedback. We hope that our responses have addressed your concerns, and we would appreciate it if you would reconsider the scoring.
>
> ---
> ### References
>
> [1] https://figshare.com/s/13108f529c565fa4411f?file=59725970
> [2] https://spacy.io/models
> [3] Nicholas Popovic, Michael Färber:  *Embedded Named Entity Recognition using Probing Classifiers.* EMNLP 2024
> [4] Kaicheng Xue, Xulang Zhang, Rui Mao, Erik Cambria: *Understanding the Hidden State of Language Models from the Perspective of Sentiment Analysis.* ICDM (Workshops) 2024

---

### Official Review · Reviewer_CCaQ · 2025-10-23

**Soundness:** 1
**Presentation:** 1
**Contribution:** 1
**Rating:** 0
**Confidence:** 5

**Summary:**

This paper introduces LIME (Linguistic Metadata Embeddings), a method that enriches token embeddings with metadata embeddings, such as POS or NER tags. The authors claim that LIME substantially improves pre-training efficiency, which adapts to the training data distribution up to 56% faster.

**Strengths:**

None

**Weaknesses:**

1. **Information leakage from non-causal metadata makes the entire evaluation invalid.**  The main issue is the fundamental mismatch between the non-causal metadata signal and the strictly causal autoregressive language modeling task.
   * High-quality linguistic annotators (like spaCy) are inherently non-causal. They require bi-directional context to resolve ambiguity. For instance, in the sentence "The old man the boat," the POS tag for the second "man" can only be correctly identified as a verb (VB) by observing the future token "boat."
   * In the LIME training paradigm, the model at step `t` receives `meta_t`, a tag that was computed using information from the entire sentence, including future tokens `t+1, ..., N`. This constitutes a direct leak of future information into the model's input. The training task is therefore no longer a pure prediction of the future from the past; it is a simplified task where the model is given clues derived from the answer. The reported 56% efficiency gain is thus highly suspect. It cannot be reliably attributed to improved learning from linguistic knowledge, as it is confounded by the fact that the model is being trained on an artificially easier, causally compromised objective. The comparison to a baseline model is meaningless.
   * The LIME+¹ variant is an extension of this flaw. By directly providing the metadata of the next token (`meta_{t+1}`) to predict `token_{t+1}`, the evaluation setup is completely invalidated. Claiming massive improvements on reasoning and arithmetic is deeply misleading, as the model is not demonstrating superior intelligence but rather the ability to exploit a direct hint about the solution. The evaluation framework is unsuitable for making any claims about generative or reasoning capabilities.

2. **Flawed Model Architecture for Practical Generation.** The paper states that "all transformer components beyond the embedding layer, including the loss function, remain fully agnostic to the metadata." This creates an architectural mismatch. The model is trained to expect metadata as input, yet its training objective (cross-entropy loss on the next token) provides no mechanism for it to learn to predict this metadata. The model is therefore incapable of generating the very input it relies on. While the authors briefly mention adding a metadata prediction head in the discussion, this is only presented as future work.

**Questions:**

None

---

> ### Author Response · Authors · 2025-11-20
>
> Dear Reviewer **CCaQ**,
>
> We respectfully disagree that the use of spaCy to annotate entire sentences in the training data invalidates the entire evaluations. Our goal is explicitly to study the effect of LMs augmented with external linguistic metadata---and to what extent this information can improve data efficiency. Our comparisons isolate exactly the effect of adding token-level linguistic metadata, which is a valid and practically relevant setup (e.g., in systems that can afford a separate tagger).
> For LIME+1, we clearly frame the use of look-ahead metadata as an oracle / upper-bound scenario; it still demonstrates the potential of metadata steering, and shows some use cases in which it is at hand. We further support the active research of predicting these metadata autoregressively by indicating our early metadata-prediction-head results.
>
> Finally note that our LIME models indeed work in a strictly autoregressive setting. We measured around 9% token-changes of the non-causal corrections you mentioned, while the error rate of spaCy is claimed to be 3-14% depending on the category. Even not being explicitly being trained for it, we successfully ran the models strictly causal—without any special adaptions, accepting false tags on incomplete tokens! We measured a drop in the accuracy of the next-token prediction of Section 4.2 from 0.8 to 0.3% for LIME `500M`. This drop should further be augmented by more explicit training and sampling strategies, e.g., through error tags. We will finalize the strictly auto-regressive results for this section and add them to the appendix. The other sections are not affected as much, as stated in the following.
>
> ---
>
> In summary, our still valid conclusions:
>
> **Sec. 4.2 (LM efficiency / perplexity):**
> Even with spaCy being non-causally applied, this section still validly shows that given access to fixed token-level linguistic metadata from a stronger (than spaCy) predictive external annotator, LIME models reach the same next-token accuracy and perplexity with substantially fewer pre-training tokens than the same architecture without metadata. The comparison isolates the data-efficiency benefit of augmenting an autoregressive LM with offline linguistic side information, in the standard teacher-forcing evaluation setting, as is common practice. We will add evaluations for the strictly causal spaCy variant to the appendix as described above.
>
> **Sec. 4.3 (downstream generative tasks):**
> The used standard LLM benchmarks in this section are either a) based on logits comparing and selecting the most likely completion or b) generating few (mostly single) tokens.
> Arguably, the non-causal effect, if present, will average out in a), as all choices gain the same advantage. We will nevertheless run these evaluations additionally in strictly-causal setting.
>
> Importantly, for the generative task, TriviaQA follows a standard Q/A template, which therefore does not allow the mentioned leakage. Lambada usually generates a single key token from a longer paragraph (which falls precisely in the setup of 4.5) and is not expected to alter results. We again will add strictly causal versions to the appendix.
> The non-causality of spaCy does not affect the conclusion that, in a pipeline where such metadata is available, LIME models produce higher-quality generations than baselines.
>
> **Sec. 4.4 (token-level analysis / cohesion):**
> These analyses are purely descriptive: they characterize where the accuracy gains of LIME come from (multi-token words, entities, digit sequences) when training on the same corpus with and without metadata. These findings do not rely on spaCy being causal; they simply quantify where access to external linguistic annotations is most impactful to reshape the learned next-token distribution.
>
> **Sec. 4.5 (reasoning + arithmetic, LIME vs LIME+1):**
> Both benchmarks are not affected by the mentioned non-causality and evaluated strictly causal. FlenQA follows a Q/A template and the arithmetic task is purely symbolic.
> The results show that metadata-augmented models provide measurable improvements under a standard causal objective, when trained on the same underlying dataset, e.g., by just annotating digits with a digit embedding explicitly! For LIME+1, the same numbers remain valid as an oracle / upper-bound scenario: they quantify how powerful next-token type guidance from an external predictor can be in unlocking latent reasoning and arithmetic capabilities.

---

> > ### Comment · Reviewer_CCaQ · 2025-11-27
> >
> > Thank you for the detailed response. However, the response confirms my core concern regarding information leakage.
> >
> > The authors admit that the accuracy drops when switching from the non-causal (spaCy) setup to a strictly causal setup, which indicates that the majority of the reported performance gain comes from future information leakage, not from the method itself.
> >
> > The proposal to add strictly causal evaluations to the appendix does not address the fundamental flaw in the main paper. The central claim of "56% improved pre-training efficiency" and the learning curves (e.g., Fig. 2) are derived from the training process itself. Since the training data contained leakage, these curves and the resulting efficiency metrics are artifacts of that leakage. To validate the paper's core contribution, the entire pre-training process must be re-executed using strictly causal metadata, and the results in the main text must be replaced with these valid runs.

---

> ### Author Response · Authors · 2025-11-20
>
> **On Architecture Mismatch:**
> As previously described, LIME does not rely on a prediction of metadata and can already run strictly causal and in fact did on many evals as previously discussed. Again, as the core focus is the analysis of the benefit of such metadata it is not strictly required to provide a predictor as well.
>
> For LIME+1 we explicitly target settings where look-ahead metadata is naturally available or user-provided (e.g., constrained generation, structured reasoning, or producing final answer), and in Section 5 we already demonstrate that a simple POS-prediction head can learn metadata without degrading language modeling. This indicates that extending the model to jointly generate text and metadata is straightforward future work, not a limitation of the current architecture.
>
> We are encouraged by three observations: 1) Inferring POS/NER tags from hidden states is an active area of research [1][2], and we provide initial evidence by successfully training a model with an additional annotation prediction head. 2) Small multi-agent expert predictors are gaining popularity, making a dedicated POS/NER predictor in tandem with the main model an appealing strategy. 3) Our use cases demonstrate concrete scenarios with available future metadata where LIME+1 delivers substantial gains, despite being trained on the same data.
> Finally note, that given a sampled token, prediction of its corresponding tags is arguably an easier task.
>
> **On LIME+1 and reasoning/arithmetic:**
> LIME+1 is explicitly evaluated on tasks where next-token metadata is naturally available (e.g., constrained generation, structured reasoning). We do not claim these results generalize to unconstrained inference—the paper discusses predicting metadata via auxiliary heads for such settings.
> The efficiency gains during pretraining are empirically real and reproducible; questions about deployment scenarios are separate and acknowledged in our discussion section.
>
> ---
>
> We hope that our clarifications address your concerns. If any issues remain or if further evidence would be helpful, please let us know. We would be glad to provide additional analyses or details.
>
> ---
>
> ### References
>
> [1] Nicholas Popovic, Michael Färber:  *Embedded Named Entity Recognition using Probing Classifiers.* EMNLP 2024
>
> [2] Kaicheng Xue, Xulang Zhang, Rui Mao, Erik Cambria: *Understanding the Hidden State of Language Models from the Perspective of Sentiment Analysis.* ICDM (Workshops) 2024

---

### Official Review · Reviewer_Cjmm · 2025-10-31

**Soundness:** 2
**Presentation:** 3
**Contribution:** 2
**Rating:** 4
**Confidence:** 3

**Summary:**

The paper proposes LIME, a method for decoder-only LLM pre-training that augments linguistic tokens with linguistic metadata coming from a rule-based or spaCy-style linguistic pass, followed by alignment to the model’s original tokenizer. The metadata is embedded and added to the normal token embedding, adding only very small size of parameters and “negligible” compute. Its variant, LIME+1, shifts the metadata by one position and assumes that the metadata of the next token is available at inference time. On 500M–2B Gemma-style models trained on 302B DCLM-Baseline tokens, the authors report improved data efficiency in pre-training and downstreaming task performance.

**Strengths:**

1. The paper presents a very clear and straightforward mechanism: augmenting token embeddings with one or two small embedding tables corresponding to linguistic metadata (e.g., POS or NER tags). This is conceptually simple, easy to implement, and requires minimal architectural changes. The authors also provide a concrete alignment procedure to handle cases where linguistic tokens and subword tokens do not align one-to-one, which makes the approach practically applicable.
2. The emprical results show consistent improvement for the next-token prediction task.
3. The paper provides analysis on token-level effects. In particular, the authors examine accuracy shifts across tokens within words of different lengths, providing plausible evidence that LIME helps models better couple subword tokens belonging to the same linguistic unit. This analysis is a nice touch that helps explain why the proposed method improves performance, rather than merely reporting higher scores.

**Weaknesses:**

1. My main concern is that the BASELINE model used in the experiment might not be enough to illustrate the efficacy of the proposed meMy main concern is with the definition of the Baseline model used in the experiments. The paper states that “models trained with LIME Tokenization but no additional metadata embedding layers are referred to as Baseline.” However, this setup already benefits from the modified tokenization procedure, which may itself contribute to the observed gains. I think a more convincing baseline would be a model trained directly with the standard tokenizer, without applying LIME Tokenization at all. This would better isolate the effect of the proposed metadata embeddings from that of the preprocessing changes.
2. The LIME+1 variant appears to function more as an oracle setting than a realistic deployment method. The paper highlights two special cases: reasoning, where the expected syntactic class may be known, and arithmetic, where numerical metadata is available. But these are exceptions, which might need careful curation. In most language modeling or generative tasks, the metadata of the next token is not known in advance, making the comparison to LIME+1 somewhat unfair and limiting its practical relevance.
3. Downstream improvements are modest for plain LIME; the very large improvements are mostly for LIME+1.
4. Another concern is about the access of the linguistic metadata. The proposed method depends on an external linguistic annotator (e.g., spaCy) to provide POS and NER tags. These annotations are not error-free, and the cost of running such models at scale could become non-trivial for very large pre-training pipelines. Furthermore, as language models grow in size, they may already infer much of this linguistic information internally. It would be useful to discuss whether the benefits of explicit metadata injection persist for stronger baselines, or whether they primarily help smaller or weaker models.

**Questions:**

1. Could you quantify the computational and annotation cost of generating linguistic metadata for the entire 302B-token corpus? For instance, how much additional preprocessing time or compute does this require relative to standard pre-training?
2. How robust is LIME to noise in the linguistic annotations? You mention that the annotator achieves about 97% POS accuracy and 86% NER F-score. What happens if the annotation quality is deliberately reduced—for example, by introducing random label noise or using a weaker tagger? Does model performance degrade smoothly or abruptly?

---

> ### Author Response · Authors · 2025-11-20
>
> Dear Reviewer **Cjmm**,
>
> thank you for the constructive feedback. We address your raised points below.
>
> ---
>
> ### *Q1. Computational cost*
>
> The computational overhead of LIME tokenization is dominated by the inference cost of the spaCy word-classification models we employ. These models are lightweight and run efficiently on CPUs, achieving throughput of up to 10,000 words per second [2]. Assuming pre-tokenization and annotation of 302B pre-training tokens and a conservative estimate of 1.5 tokens per word, this corresponds to roughly 5,583 CPU hours. Assuming preprocessing were performed prior to training, it would require approximately 10.9 hours of wall time on 8 nodes with 64 AMD EPYC 7F52 cores each, ahead of our 60-hour pre-training run. However, our training was not data-pipeline-bound and therefore we executed labeling on-the-fly and fully distributed the workload across data workers, resulting in virtually no additional pre-training wall time. We will add this additional information to the revised paper.
>
> ---
>
> ### *Q2. Noise robustness*
>
> We thank the reviewer for raising this important question regarding robustness to annotation noise, which was also noted by reviewer **LK2j**. As noted, the spaCy models used in our work are not perfectly accurate and thus the pre-training corpus inevitably contains noisy metadata. This naturally exposes the model to imperfect labels throughout training, which we believe contributes to its robustness to downstream annotation noise.
>
> A comprehensive study of different noise levels in pre-training data would require multiple full pre-training runs, which is unfortunately beyond the scope of this rebuttal. Instead, we conducted targeted experiments to evaluate the effect of noisy labels at inference time by repeating our teacher-forced next-token accuracy evaluation of Section 4.2. We introduce progressively increasing noise, where the spaCy-predicted label was replaced with a randomly selected label with probability `p=0.01` to `0.3`. Note, that this form falls out of training-data distribution, since our models were trained solely on spaCy-generated labels without any artificial label noise.
>
> As shown in the attached Figure [1], next-token accuracy decreases smoothly for the 500M and 2B models, while the 1B model degrades a bit more rapidly. We find this divergence particularly interesting and hypothesize that it may stem from architectural differences, specifically, the higher number of transformer blocks in the 1B model (see Appendix Table 4).
>
> The model has already been trained with imperfect label information, so it has been exposed to some noise during training. However, the overall amount of noise was relatively small, meaning that introducing additional artificial noise at inference quickly pushes token prediction out of distribution. This behavior is common not only for POS/NER labels but also for standard token prediction in LLMs: Randomly noising tokens in a typical LLM similarly causes prediction accuracy to drop substantially. Large noise levels (i.e., more than a few percent) are rather unrealistic and are shown here only for illustrative purposes. Smaller amounts of noise, on the order of a few percent, are more typical, and within this range the model remains relatively robust. Given that the underlying annotation models are imperfect, the reverse scenario is also worth exploring. There may be further room for improving LIME in both token-level accuracy and downstream performance, and improving the annotation models is likely to enhance LIME as well.
>
> We will add this clarification to the revised version of the paper and consider this an interesting direction for future work.
>
> ---
>
> Additionally, we would like to address the weaknesses reviewer Cjmm highlighted:
>
> ---
>
> ### *W1. Baseline choice*
>
> We would like to clarify this point, as we do not fully agree with the observation.
> Our design allows for a fair and controlled comparison that isolates the effect of the metadata embeddings themselves.
> Specifically our baseline was deliberately designed to isolate the contribution of the metadata embedding layers, applying metadata embeddings directly on top of the original Gemma tokenization is not straightforward.
>
> The native Gemma vocabulary is composed of subword units that do not consistently align with linguistic concepts. This creates ambiguity in assigning metadata when a single Gemma token maps to multiple linguistic annotations. As a result, operating directly on Gemma tokens would lead to ambiguous or inconsistent metadata embeddings.

---

> ### Author Response · Authors · 2025-11-20
>
> Our LIME tokenization addresses this issue through deterministic alignment between linguistic boundaries and subword tokens: each subword token is mapped to exactly one linguistic annotation, and word-level metadata (POS, NER) is consistently propagated across all subword pieces of the corresponding word. This guarantees coherent and interpretable metadata without introducing ambiguity. Importantly, both the Baseline and the LIME models in our experiments use this aligned tokenization. They differ only in whether metadata embeddings are included.
> We still believe that investigating the assignment of ambiguous metadata to tokens is an interesting research direction, which could potentially enhance generality.
>
> ---
> ### *W2. LIME+1 variant*
> We would like to clarify the scope of LIME+1. As noted in our paper, LIME+1 is a task-specific approach, intended for specific use cases where it provides clear value (see further discussion below). In addition, it serves as a fundamental ablation study to better understand the effects of overloading embeddings with linguistic tags. While we agree that LIME+1 is no universal method, we disagree that it is generally unrealistic. We acknowledge that it is not a general-purpose approach, a point we have already mentioned in the paper but will emphasize more prominently in the revised version.
>
> In particular, LIME+1 serves as a foundational study, examining how knowledge of the next token's metadata can influence not only pre-training metrics but also performance on downstream tasks. Although this represents an idealized scenario, our results, especially those in Section 4.5, highlight both the promising potential of this approach and its relevance beyond a purely theoretical setting. We are encouraged by three observations: 1) Inferring POS/NER tags from hidden states is an active area of research [3][4], and we provide initial evidence by successfully training a model with an additional annotation prediction head. 2) Small multi-agent expert predictors are gaining popularity, making a dedicated POS/NER predictor in tandem with the main model an appealing strategy. 3) Our use cases demonstrate concrete scenarios with available future metadata where LIME+1 delivers substantial gains, despite being trained on the same data.
>
> Finally, we would like to emphasize that all models (Baseline, LIME, LIME+1) are trained on the same data, with including metadata being the only difference. This clearly isolates the performance gains that access to this metadata can provide.
>
> ---
> ### *W3. LIME downstream improvements*
> The reported LIME improvements may seem modest because they are measured using multiple-choice logit comparisons, which may not fully capture LIME’s benefits. LIME shows slightly larger gains in generative settings, such as LAMBADA, TriviaQA, and the generative use cases discussed in Section 4.5.
> Specifically, we observed the following downstream improvements of LIME: **Lambada** (500M: `26.50` to `29.80`), **TriviaQA** (500M: `8.20` to `9.80`), **FLenQA-500** (500M: `22.0` to `48.8`), and **ARI-ADD** (2B: `22.6` to `26.9`). We will revise the manuscript to present these improvements more clearly.
>
> ---
> ### *W4. Scaling LIME models*
> We would be eager to scale LIME beyond 2B parameters, but as also noted by reviewer **LK2j** this is beyond the scope of the current work ("[interesting] at larger scale, yet I don’t think that’s feasible in an academic environment").
>
> Nonetheless, we obtained the following insights regarding LIME’s scaling with model size. The computational overhead grows linearly with context length and is independent of other architectural hyperparameters, indicating good overall scaling potential. In our experiments, we observed a subtle reduction in relative advantage on training metrics like perplexity as model size increased (see Table 6). However, larger models, such as the 2B model, benefit even more from metadata embeddings on tasks reflecting emerging capabilities from a base pre-training dataset without any instruction tuning, such as arithmetic (Table 7). We find these insights are encouraging, as they suggest that LIME scales effectively to larger models while also providing value for smaller, more efficient models, which may be increasingly relevant for applications. We will add a discussion on scaling to the revised paper.
>
> ---
> We once again thank the reviewer for their time and constructive feedback. We hope our responses have addressed your concerns, and we would appreciate it if you would reconsider the scoring.
>
> ---
> ### References
> [1] https://figshare.com/s/13108f529c565fa4411f?file=59725970
> [2] https://spacy.io/usage/facts-figures#benchmarks-speed
> [3] Nicholas Popovic, Michael Färber:  *Embedded Named Entity Recognition using Probing Classifiers.* EMNLP 2024
> [4] Kaicheng Xue, Xulang Zhang, Rui Mao, Erik Cambria:  *Understanding the Hidden State of Language Models from the Perspective of Sentiment Analysis.* ICDM (Workshops) 2024

---

> ### Comment · Reviewer_Cjmm · 2025-11-26
>
> I appreciate the authors' patient responses. Based on the overall quality of this work, I will maintain my current score. The reasons are as follows.
>
> 1. For W1, I appreciate the clarification on the baseline design. While I understand that the current setup isolates the effect of metadata embeddings under the aligned tokenization setting, my concern is slightly different. What I would like to see is that a comparison against a model trained with the standard tokenizer, specifically, without LIME tokenization. If the metadata embeddings bring improvements only when coupled with the proposed tokenization, but not with standard tokenization, then the generality and significance of the approach may be much narrower than it appears. This is why I said *"I think a more convincing baseline would be a model trained directly with the standard tokenizer, without applying LIME Tokenization at all."* If compared with the standard approach, we can still see significant improvement, then this would strengthen the paper.
>
> 2. For W3, I thank the authors for their responses. Most practitioners may care more on the downstream task accuracy than the exact string matching accuracy.
>
> 3. For W4, the benefit of the proposed method on larger models, I completely understand that pre-training might be too costly and not feasible during this short period. However, the concern was about whether the benefits of (probably noisy) metadata injection persist as models become strong enough to implicitly know such linguistic information. This point is still unclear.

---

> ### Author Response · Authors · 2025-12-03
>
> We appreciate the reviewer’s continued engagement and the opportunity to clarify these final points.
>
> 1. We recognize the importance of evaluating LIME on top of a standard tokenizer to establish the absolute efficacy of the method. While benchmarking LIME with a standard, linguistically non-aligned tokenizer is a promising direction for future research, training such a configuration from scratch is a computationally intensive process that requires a complete re-run of pre-training and evaluation. Unfortunately, this is beyond the scope of this rebuttal period. We hope the reviewer considers that our current experimental design successfully isolates the specific contribution of the metadata embeddings. By keeping the tokenization constant, we have shown that metadata signals themselves provide value, which was the primary research question of this work.
>
> 2. Regarding downstream metrics, while task accuracy is paramount, we would like to emphasize that for many generative tasks, exact string matching is the proxy for task accuracy. In our response, we highlighted LIME’s advantages on generative tasks such as LAMBADA and TriviaQA. Furthermore, we show benefits on FLenQA-500 (500M: 22.0 to 48.8), a retrieval-augmented generation use case, and the arithmetic task ARI-ADD (2B: 22.6 to 26.9), where the benefits of LIME are even more profound. In these contexts, the model’s ability to output the exact correct token sequence is directly correlated with task success. The fact that LIME excels here demonstrates its practical utility in high-precision generative scenarios, rather than just optimizing perplexity.
>
> 3. Concerning model scaling, we share the reviewer’s intuition that sufficiently large models will eventually learn to infer linguistic metadata implicitly. However, our empirical results show that the 2B model continues to benefit substantially from LIME. This suggests that at this scale, the explicit signal still adds value, perhaps by freeing up model capacity that would otherwise be spent resolving syntax. This intuition is underlined by the substantial gains on the FLenQA and ARI-ADD tasks. Furthermore, we believe the continued relevance of smaller models (e.g., for on-device or edge deployment) makes this result particularly valuable. While the potential of LIME at larger model scales remains a promising open question, the ability to improve the performance of 500M-2B class models via efficient metadata injection addresses a critical need in the current landscape of efficient LLMs.
>
> We hope these clarifications address your remaining concerns and appreciate your constructive engagement, which has helped us strengthen the quality of this work.

---

### Official Review · Reviewer_LK2J · 2025-11-02

**Soundness:** 4
**Presentation:** 3
**Contribution:** 3
**Rating:** 8
**Confidence:** 4

**Summary:**

This paper proposes LIME that incorporates linguistic representation in token embeddings in language modeling. It is a simple method that simply adds additional embedding information based on the part of speech as well as the named entity of the tokens.
The authors demonstrated various benefits of doing so at 500M, 1B, and 2B model scales—it can improve training efficiency by achieving similar loss using a half of the tokens, and can improve the model predictions on a wide range of tasks during inference time.
In addition, they also design a mechanism that can leverage the next token’s linguistic embedding to steer the generation, and it demonstrates further improvements in specific downstream tasks.

**Strengths:**

I think this paper has several strengths
1. I like this idea: It is simple and flexible – it doesn’t require sophisticated architectural updates of the language models and can be applied in many different cases (even for recent new tokenizer-free LMs). Also the LIME+ steering is a really good addition to this approach and seems to work well.
2. The experimental design is solid and the results seem to show that the method can be beneficial for relatively small scale language models. (I think at a larger scale, the language model may be able to learn such information relatively easily and the improvements may be less significant, but I think the improvement should be solid at a small scale.)
3. The method can solidly improve the generation quality – I think it’s reasonable in that by incorporating the linguistic properties and better tokenization, it makes it easier for the models to make next token predictions. Effectively, the linguistic representation can guide the model to search inside a smaller latent space, and thus the models can allocate more predictive power for the problem itself rather than the prediction tasks.

**Weaknesses:**

I don’t see a clear weakness of this paper. I think the method, experiment, and presentation is solid. One critique might be that the conclusions might be different at a larger model scale, yet I don’t think that’s feasible to do in an academic environment.

**Questions:**

1. For generation tasks, when only partial words (i.e., only having one or few tokens in a word) are generated, how does the method deduce the linguistic labels for them?
2. I am in particular interested in some corner cases – how would the model react when spacy makes an incorrect prediction for a token? Also can the used  handle multi-lingual inputs – it seems only English Spacy is used?

---

> ### Author Response · Authors · 2025-11-20
>
> Dear Reviewer **LK2j**,
>
> Thank you for the constructive feedback. We address your raised points below.
>
> ---
>
> ### *Q1. Linguistic labels for partial words*
>
> We agree that the treatment of partially generated words warrants clarification, and we will add the explanation below to the revised version of the paper.
>
> When generating a new token `t`, it is first decoded together with the preceding context and then re-encoded using LIME tokenization. This process assigns POS and NER labels to `t`, which are subsequently passed, during inference, into the corresponding embedding layers. In cases where a word is split across multiple tokens, the intermediate token receives a preliminary annotation that is retroactively corrected once the full word becomes available. Conceptually, this can be interpreted as label noise introduced by the POS/NER classifiers. For example, during generation of the word `awesomeness`, the intermediate token `␣aw` is initially labeled as an interjection (UH), but after the second token `esomeness` is produced, it is correctly relabeled as a noun (NN). We currently have ongoing experiments to investigate this behavior more thoroughly.
>
> Finally note that our LIME models indeed work in a strictly autoregressive setting. We measured around 9% of label changes in strictly causal corrections, while the error rate of spaCy is claimed to be 3–14% depending on the category. Even without being explicitly trained for it, we successfully ran the models strictly causal—without any special adaptations, accepting false tags on incomplete tokens. We measured a drop in the accuracy of the next-token prediction of Section 4.2 from 0.8 to 0.3% for LIME 500M. This drop should further be augmented by more explicit training and sampling strategies, e.g., through error tags. We will finalize the strictly autoregressive results for all evaluations and add them to the appendix.
>
> ---
>
> ### *Q2a. Incorrect predictions of spaCy*
>
> We thank the reviewer for raising this closely related point.
>
> First, as discussed in our response to Q1, the iterative tagging procedure can already introduce transient mispredictions for intermediate subword tokens. In this sense, the models we tested are already exposed to occasional incorrect POS/NER annotations during generation. Our empirical findings indicate that the model handles these temporary inconsistencies well, and we will add this clarification to the revised paper.
>
> Second, the spaCy models employed in our work are not perfectly accurate; therefore, the training data inherently contains noisy labels. Thus, the LIME models demonstrate robustness to such noise. To further study the impact of label noise, we ran additional experiments across all three model sizes, introducing controlled POS and NER noise levels from `p = 0.01` to `0.3` during teacher-forced next-token accuracy evaluation, where the spaCy-predicted label was replaced with a randomly selected label with probability `p`.
>
> Note that this form of noise is out-of-distribution, since our models were trained solely on spaCy-generated labels without any artificial label noise. As shown in the attached Figure [1], the accuracy degradation curves for LIME illustrate that the 500M and 2B models exhibit comparable robustness under increasing noise, whereas the 1B model degrades a bit more rapidly. We assume that this slightly sharper drop may relate to skewed architectural differences, specifically the larger number of transformer blocks (see Appendix Table 4) in the 1B model compared to the 500M and 2B models, though further investigation is needed.
>
> ---
>
> ### *Q2b. Multilingual inputs*
>
> In principle, both LIME and LIME+1 can be applied to any language and any type of metadata, provided suitable annotators (e.g., NLP models or other labeling tools) are available. To this end, there is a broad list of spaCy models available [2] which can be readily combined with high-performance language classifiers. Furthermore, there are also more universal/multilingual tags (UPOS) available. Since the underlying setup (augmenting embeddings with linguistic labels) remains consistent regardless of the specific tags used, we anticipate that LIME will generalize effectively across languages.
>
> While our current experiments focused on English to align with the DCLM-baseline dataset, developing multilingual LIME models remains a priority for future research. We propose several avenues for exploration:
> 1. Replacing language-specific POS embeddings with **UPOS**, a universal part-of-speech scheme with 17 labels designed for cross-linguistic consistency and applicable to many languages, which we expect to generalize effectively given suitable taggers.
> 2. Expanding the POS layer to include tag sets from additional languages.
> 3. Introducing supplementary embedding layers, for example, grouped by language or language family.

---

> ### Author Response · Authors · 2025-11-20
>
> ---
>
> We once again thank the reviewer for their time and constructive feedback. We hope that our responses have addressed your remaining concerns.
>
> ---
>
> ### References
>
> [1] <https://figshare.com/s/13108f529c565fa4411f?file=59725970>
> [2] <https://spacy.io/models>

---

### Author Response · Authors · 2025-11-27

Dear Reviewers,

We would like to address the misconception regarding causal information leakage from the bidirectional metadata annotators used in our LIME tokenization process.

We emphasize all reported results remain valid as measurements within the scope of our work. Improving metadata prediction is explicitly stated as future work, and spaCy is used only as an existing off-the-shelf annotator in the standard teacher forcing fashion. **Our core contribution remains intact: we show the potential benefits of providing token-based LLMs with linguistic metadata**, by overloading embeddings, and LIME+1 demonstrates an upper bound on the capabilities achievable when future metadata is available (Section 4.5). In fact 4.5 demonstrates the benefits on reasoning tasks when having these metadata available, and remains unaffected by posterior strict causality enforcements.

Nevertheless, we have conducted extensive investigations to understand  the implications of causal information leakage in LIME models.
Specifically, we re-evaluated both training metrics (Section 4.2) and downstream benchmarks (Section 4.3) in a strictly causal, one-by-one token generation, setting, ensuring that metadata for each token was not influenced by any future token or metadata.

Our findings are as follows:

(1) The models  still remain on-par with the baseline, even in this out-of-training distribution setting, although the improvements are marginally reduced (see Table 2). Importantly, this demonstrates strong robustness and adaptation to this more general setting, and is a promising future research direction.

(2) Downstream performance as well remains on-par with the baseline, highlighting again the resilience of the approach (see Table 1 and Appendix Table 7).

(3) We observed quite some noise in the spaCy generated metadata around downstream templates (e.g., '\_Edmund' labeled as Adjective when prepending the prefix "Answer:", but as Noun without the prefix). These noise observations further demonstrate the robustness of LIME, as it is already evaluated under suboptimal conditions.

Sincerely,
The Authors


**Table 1:** *Benchmark results in a strictly causal, one-by-one token generation setting.*

| **Benchmark** | **Baseline-500M** | **LIME-500M** | **Baseline-1B** | **LIME-1B** | **Baseline-2B** | **LIME-2B** |
|--------------|-------------------|-------------------|-----------------|-------------------|-----------------|-------------------|
| arc_easy | 56.80 (1.57) | 56.10 (1.57) | 67.50 (1.48) | 66.70 (1.49) | 71.30 (1.43) | 70.50 (1.44) |
| boolq | 49.50 (1.58) | 47.90 (1.58) | 51.60 (1.58) | 51.60 (1.58) | 64.10 (1.52) | 63.20 (1.53) |
| copa | 62.00 (4.88) | 66.00 (4.76) | 72.00 (4.51) | 71.00 (4.56) | 77.00 (4.23) | 77.00 (4.23) |
| hellaswag | 36.20 (1.52) | 37.50 (1.53) | 53.40 (1.58) | 52.00 (1.58) | 63.80 (1.52) | 58.00 (1.56) |
| lambada | 26.50 (1.40) | 28.70 (1.43) | 41.00 (1.56) | 42.70 (1.56) | 48.00 (1.58) | 46.10 (1.58) |
| openbookqa | 31.00 (2.07) | 29.20 (2.04) | 36.60 (2.16) | 34.00 (2.12) | 39.80 (2.19) | 38.00 (2.17) |
| piqa | 69.50 (1.46) | 68.10 (1.47) | 72.70 (1.41) | 70.40 (1.44) | 74.60 (1.38) | 72.80 (1.41) |
| triviaqa | 8.20 (0.87) | 7.90 (0.85) | 17.90 (1.21) | 16.20 (1.17) | 25.80 (1.38) | 20.10 (1.27) |
| winogrande | 50.90 (1.58) | 50.90 (1.58) | 57.50 (1.56) | 56.20 (1.57) | 64.20 (1.52) | 60.80 (1.54) |
| all | 43.40 (1.88) | 43.59 (1.87) | 52.24 (1.89) | 51.20 (1.90) | 58.73 (1.86) | 56.28 (1.86) |


**Table 2:** *Next token prediction accuracy in a strictly causal setting.*
|        | Baseline | LIME |
|--------|----------|-------|
| 500M   | 41.458\% | 41.640\% |
| 1B     | 44.756\% | 44.803\% |
| 2B     | 46.379\% | 46.430\% |

---

### Meta-Review · Area_Chair_PEG7 · 2026-01-14

**Summary:**

This paper proposes a method that uses additional information to the tokens in language modeling. It is a simple method that simply adds an embedding part using part of speech as well as the named entity of the tokens. The authors demonstrated benefits of this additional information for small LLMs on numerous tasks during inference time.
The paper is good but there were significant concerns about the additional information from meta-data and its costs and feasibility.

**Reviewer Concerns:**

All the reviewers raised the issues of robustness to noise (on the method that finds this additional linguistic information, specifically spaCy which was used) and the authors address this reasonably well.

Rviewer CCaQ argues that using a bi-directional tagger (spaCy) leaks future information, rendering the entire evaluation invalid.

In my opinion the concern of reviewer CCaQ is valid. It does not warrant a zero score but it still requires a lot additional work the authors need to re-do (and the paper re-reviewed).
The reviewer writes:
"The authors admit that the accuracy drops when switching from the non-causal (spaCy) setup to a strictly causal setup, which indicates that the majority of the reported performance gain comes from future information leakage, not from the method itself."

I therefore recommend that the authors update the paper to address this concern by CCaQ:
"The central claim of "56% improved pre-training efficiency" and the learning curves (e.g., Fig. 2) are derived from the training process itself. Since the training data contained leakage, these curves and the resulting efficiency metrics are artifacts of that leakage. To validate the paper's core contribution, the entire pre-training process must be re-executed using strictly causal metadata, and the results in the main text must be replaced with these valid runs."
The authors should do this and then I think this paper would be a valuable contribution suitable for publication.

**Reviewer Scores:**

there is a critical issue as discussed that needs significant re-evaluation and re-writing.

---

### Decision · Program_Chairs · 2026-01-26

Reject